# FlpStop, a tool for conditional gene control in *Drosophila*

Yvette E Fisher[†‡], Helen H Yang[†], Jesse Isaacman-Beck, Marjorie Xie, Daryl M Gohl[§], Thomas R Clandinin*

Department of Neurobiology, Stanford University, Stanford, United States

**Abstract** Manipulating gene function cell type-specifically is a common experimental goal in *Drosophila* research and has been central to studies of neural development, circuit computation, and behavior. However, current cell type-specific gene disruption techniques in flies often reduce gene activity incompletely or rely on cell division. Here we describe FlpStop, a generalizable tool for conditional gene disruption and rescue in post-mitotic cells. In proof-of-principle experiments, we manipulated *apterous*, a regulator of wing development. Next, we produced conditional null alleles of *Glutamic acid decarboxylase 1* (*Gad1*) and *Resistant to dieldrin* (*Rdl*), genes vital for GABAergic neurotransmission, as well as *cacophony* (*cac*) and *paralytic* (*para*), voltage-gated ion channels central to neuronal excitability. To demonstrate the utility of this approach, we manipulated *cac* in a specific visual interneuron type and discovered differential regulation of calcium signals across subcellular compartments. Thus, FlpStop will facilitate investigations into the interactions between genes, circuits, and computation.

**\*For correspondence:** trc@ stanford.edu

[†]These authors contributed equally to this work

**Present address:** [‡]Department of Neurobiology, Harvard Medical School, Boston, United States; [§]University of Minnesota Genomics Center, Minneapolis, United States

**Competing interests:** The authors declare that no competing interests exist.

## Introduction

A neuron's pattern of gene expression ultimately defines its morphology, connectivity, and physiology. Most genes have multiple temporally and spatially distinct roles in different cells and can regulate both circuit development and function. Thus, to dissect the links between genes, computation, and behavior in the adult brain, gene activity must be manipulated selectively in mature, differentiated neurons in a cell type-specific manner. Here we describe a generalizable approach for completely disrupting target genes in cell populations of interest and demonstrate how this method can be used to illuminate the molecular basis of neural computation.

The fruit fly *Drosophila melanogaster* is a prominent model system in which to explore neural circuit development and function. Studies in this animal have provided crucial insights into cell fate determination, wiring specificity, and the circuit bases of sensory processing and behavior (*Bellen et al., 2010*; *Evans and Bashaw, 2010*; *Kaneko and Ye, 2015*; *Guven-Ozkan and Davis, 2014*; *Silies et al., 2014*; *Wilson, 2013*; *Yamamoto and Koganezawa, 2013*). *Drosophila* is well suited for studies of gene function in the nervous system, as its stereotyped circuit architecture allows repeated access to defined circuit components. In addition, many key neural genes are either uniquely encoded in the genome or have a small number of paralogs (*Littleton and Ganetzky, 2000*). As a result, these genes are less likely to be functionally redundant, creating a unique opportunity for defining their contributions. Finally, large collections of genetic tools place heterologous transcription factors like Gal4 under the control of genomic enhancers, enabling access to many specific cell populations (*Venken et al., 2011b*).

Understanding how specific genes determine the synaptic and intrinsic processes that implement neural computation is a critical goal. Every neuron synthesizes a specific complement of neurotransmitters and receptors that shapes synaptic communication, enabling diverse computations. For instance, in both mammals and insects, inhibitory GABAergic signaling implements signal

transformations such as gain control and normalization (*Carandini and Heeger, 1994*; *Lee et al., 2012*; *Olsen and Wilson, 2008*; *Olsen et al., 2010, 2012*; *Root et al., 2008*). During neuronal computation, these extrinsic signals interact with the intrinsic membrane properties of each neuron, which are governed by the complement of ion channels the neuron expresses. For example, a suite of voltage-gated calcium channels with different kinetics and activation thresholds mediate neuronal excitability, presynaptic vesicle release, and activity-dependent changes in transcription (*Catterall, 2011*). The subcellular distribution and biophysical properties of such ion channels have profound impacts on the dynamics of synaptic transmission and on the integration of synaptic inputs (*Abbott and Regehr, 2004*; *Migliore and Shepherd, 2002*). However, how these active conductances contribute to the encoding of information in vivo is not fully understood. The functions of voltage-gated channels have been studied extensively at the *Drosophila* neuromuscular junction. There, the calcium channel encoded by the gene *cacophony* is required for evoked transmission, suggesting that it mediates the influx of calcium that drives synaptic vesicle fusion (*Kawasaki et al., 2000, 2002*; *Kawasaki and Zou, 2004*). *Drosophila* has two additional voltage-gated calcium channels, *CaAlpha1D* and *CaAlpha1T*, which have been suggested to also play roles in neuronal excitability (*Iniguez et al., 2013*; *Worrell and Levine, 2008*; *Ryglewski et al., 2012*). However, it is incompletely understood how these channels contribute to signaling and computation in the central nervous system.

Since each of these genes has widespread functions in the brain, the ability to manipulate them in individual classes of neurons is essential. Current techniques for cell type-specific gene manipulation in *Drosophila* are RNA interference (RNAi), targeted degradation of GFP-tagged proteins (deGradFP), recombinase-mediated exon deletion, induction of somatic mutations with CRISPR/Cas9, and clonal analyses using approaches such as Mosaic Analysis with a Repressible Cell Marker (MARCM) (*Dietzl et al., 2007*; *Hakeda-Suzuki et al., 2011*; *Xue et al., 2014*; *Lee and Luo, 1999*). While powerful, these methods have limitations that restrict their utility. RNAi uses a short hairpin RNA to trigger sequence-specific gene knock down; expression of these hairpins with the Gal4 system confers cell type-specificity (*Fire et al., 1998*; *Kennerdell and Carthew, 2000*). Large RNAi libraries targeting most genes in the genome are publically available (*Dietzl et al., 2007*; *Ni et al., 2009*). However, these hairpins can have off-target effects, interfering with genes other than the gene of interest (*Ma et al., 2006*). In addition, knock down of gene expression is rarely complete and varies depending on the level of Gal4 expression. deGradFP uses cell type-specific expression of an anti-GFP nanobody to target GFP-tagged proteins for proteosomal degradation (*Caussinus et al., 2012*). However, this approach requires that GFP-tagging preserves protein function, and, similar to RNAi, the extent of gene knock down also depends on the level of Gal4 expression. Flanking critical exons of a gene with recombinase sites produces conditional alleles such that cell type-specific expression of the recombinase enables targeted gene disruption; however, this approach has seen only limited application in flies (*Hakeda-Suzuki et al., 2011*; *Choi et al., 2009*). CRISPR/Cas9 uses a small guide RNA and the Cas9 nuclease to produce small insertion and deletion mutations in the targeted gene; Gal4-driven expression of Cas9 confers cell type-specificity. However, somatic expression of Cas9 produces different mutations in each cell and has not been shown to work in fly neurons. Critically, RNAi, recombinase-mediated deletion, and somatic CRISPR/Cas9 all lack a cell-by-cell indicator of gene disruption, an essential feature for cell type-specific manipulations. In contrast, methods such as MARCM use mitotic recombination to create labeled homozygous mutant cells within an otherwise heterozygous animal. However, by design, MARCM is incompatible with Gal4 driver lines that are only expressed postmitotically, precluding selective manipulation of all cells of a single type in the adult fly.

Here we report FlpStop, a tool that can be used in differentiated and undifferentiated cells to produce either complete gene disruption or rescue. By design, FlpStop is specific for both the target gene and cell type and provides a cell-by-cell readout of the manipulation. FlpStop uses insertional mutagenesis to create conditional null alleles and reports gene manipulation through positive labeling of targeted cells. We tested this approach in eight different genes that are involved in development, neuronal signaling, and intrinsic excitability—*apterous* (*ap*), *cacophony* (*cac*), *Choline acetyltransferase* (*ChAT*), *Glutamic acid decarboxylase 1* (*Gad1*), *paralytic* (*para*), *Resistant to dieldrin* (*Rdl*), *Shaker cognate l* (*Shal*), and *Vesicular glutamate transporter* (*VGlut*)—and generated five conditional null alleles and two conditional hypomorphs. We demonstrate that FlpStop can efficiently impair gene function within genetically targeted populations of interest and at specific times during

development. Finally, we used FlpStop in combination with in vivo calcium imaging to investigate the role of the voltage-gated calcium channel Cac in a visual interneuron. Strikingly, cell type-specific removal of Cac selectively altered visually evoked calcium signals in individual neuronal compartments in unexpected ways. While the loss of Cac reduced responses in some but not all axonal compartments, it also dramatically increased responses in a specific dendritic region. Thus, FlpStop is a powerful tool for investigating how individual genes contribute to neuronal function in vivo.

## Results

### FlpStop: a new tool for conditional gene disruption

We designed a generalizable transgenic tool that allows conditional, cell type-specific disruption of gene function. The Gene Disruption Project has produced a publically available collection of genomic insertions that can be exchanged for other DNA sequences using φC31 integrase (*Figure 1a*). These MiMIC cassettes target 2854 coding introns within 1862 distinct genes (*Venken et al., 2011a*; *Nagarkar-Jaiswal et al., 2015b*). We reasoned that a strategy that uses intronic insertions to conditionally disrupt gene function would be broadly useful. Therefore, we designed FlpStop, a small construct for conditional gene disruption capable of integrating into MiMIC insertions (*Figure 1b*). Our approach incorporated two parallel strategies to disrupt gene expression. First, the construct acts on transcription using the SV40 and Tubα1 transcriptional terminators (*Stockinger et al., 2005*) (*Figure 1a* and *Figure 1—figure supplement 1a*). Second, the construct acts on translation by incorporating the MHC intron 18 splice acceptor followed by stop codons in all three reading frames (*Venken et al., 2011a*; *Hodges and Bernstein, 1992*) (*Figure 1a* and *Figure 1—figure supplement 1a*). Similar constructs have been used previously in flies to produce strong loss-of-function alleles (*Lukacsovich et al., 2001*; *Schuldiner et al., 2008*). These disruptive elements, hereafter designated SA-STOP for brevity, are flanked by two pairs of Flp recombinase target (FRT) sites that form a FLEx-switch, making this DNA region invertible and thus conditional (*Schnütgen et al., 2003*). The FRT sites are arranged such that in the presence of Flp recombinase, the construct can be inverted and then stably locked in the opposite orientation (*Schnütgen et al., 2003*) (*Figure 1c*). Therefore, in one orientation, the splice acceptor and the transcriptional terminators of the SA-STOP cassette lie on the non-transcribed DNA strand and do not interfere with the expression (non-disrupting, ND); after inversion, the SA-STOP cassette is relocated to the transcribed DNA strand, enabling it to interfere with gene activity (disrupting locked orientation, D lock) (*Figure 1c*). A conceptually similar construct was sufficient to conditionally disrupt a mitochondrial RNA helicase in zebrafish (*Ni et al., 2012*). Using the FlpStop approach, homozygous mutant cells can be induced in otherwise heterozygous animals by placing the conditional FlpStop allele in trans to an existing null allele and expressing Flp recombinase within target cells of interest (*Figure 1d* and *Figure 1—figure supplement 1b, c*). Use of an independent null allele avoids the potential effects of off-target mutations in a genetic background, as is standard practice in the field. In *Drosophila*, precise control of Flp expression can be easily achieved by expressing Flp under either the direct control of a specific promoter or the indirect control of a driver (e.g. the Gal4 system).

Because the MiMIC cassette contains two attP target sites, two independent FlpStop insertions can be isolated at each locus: one where the initial cassette is in the non-disrupting orientation (ND, *Figure 1c*) and another where the cassette is inserted in the disrupting orientation (D, not shown). As a result, FlpStop reagents for both conditional disruption (non-disrupting to disrupting locked) and rescue (disrupting to non-disrupting locked) can be produced in a single round of transgenesis. In addition, these disrupting alleles produced by transgenesis provide a matched control in an isogenic genetic background for testing cassette mutagenicity in each locus.

To provide a cell-autonomous report of cassette inversion, we incorporated the cytosolic red fluorescent protein tdTomato into the cassette (*Shaner et al., 2004*). To mark cells in which the cassette has been inverted, tdTomato was made part of the invertible FLEx switch region (*Figure 1d* and *Figure 1—figure supplement 1a*). Upon cassette inversion, the tdTomato coding region is brought into proximity of a UAS sequence that is located just outside of the FLEx switch, thus coming under the transcriptional control of Gal4 (*Brand and Perrimon, 1993*; *Shaner et al., 2004*). This fluorescent reporter provides a direct readout of the cells in which the cassette is inverted and allows assessment of inversion efficiency under different experimental conditions.

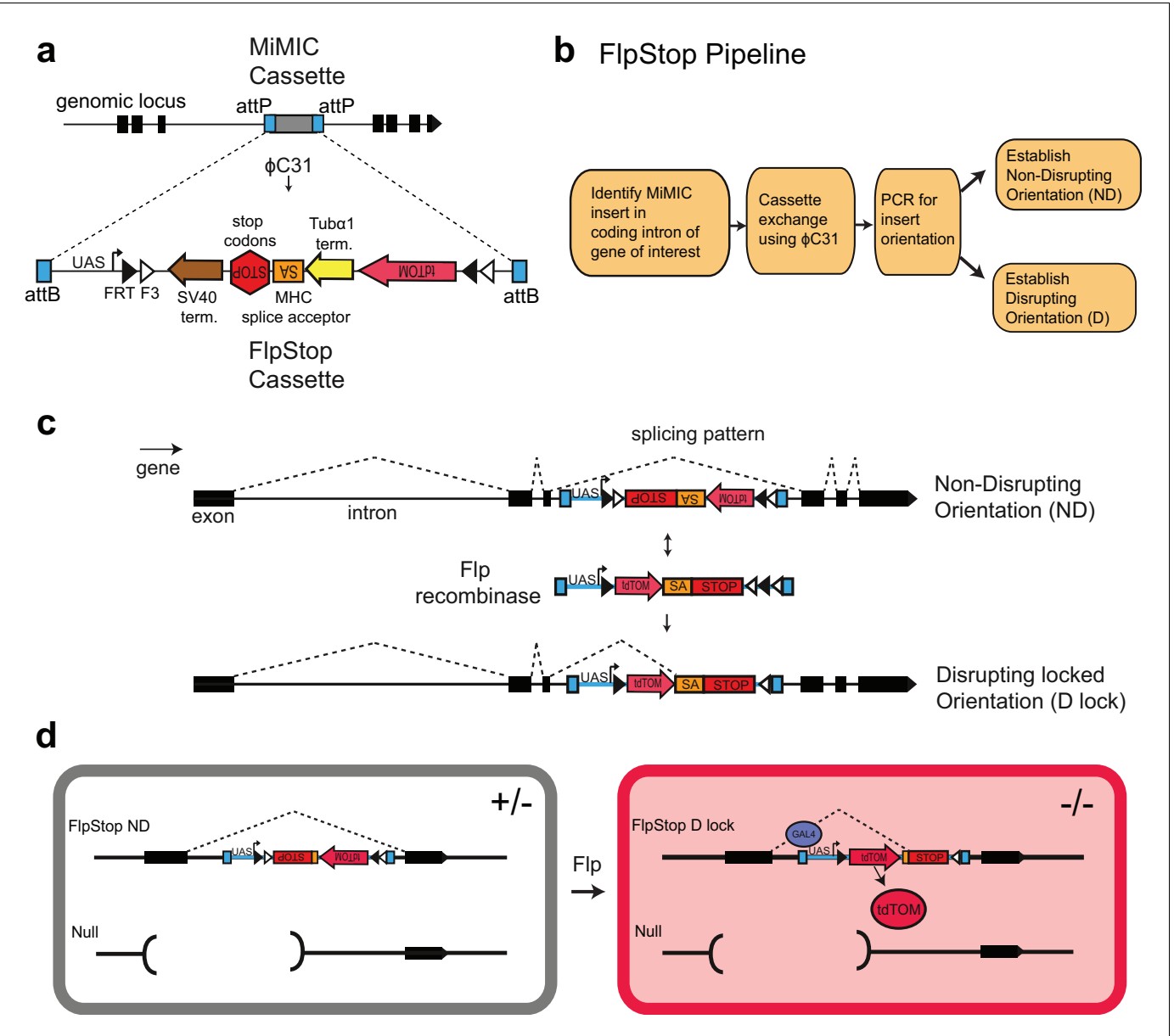

**Figure 1.** FlpStop is a generalizable tool for cell type-specific disruption of endogenous gene function. (**a**) Schematic of the internal elements of the FlpStop cassette and of how FlpStop transgenic alleles are created. An intronic MiMIC cassette (*Venken et al., 2011a*) is replaced with the FlpStop cassette using φC31. The FlpStop cassette contains the following disruptive elements: two transcriptional terminators, a MHC splice acceptor, and stop codons in all three reading frames. These disruptive elements are flanked by a series of Flp recombinase sites (FRT and F3). An upstream activation sequence (UAS) is located outside of the FRT sites and a tdTomato is encoded internal to the FRT sites. (**b**) Pipeline to create FlpStop alleles. First, a MiMIC line is identified that contains an insertion within a coding intron of a gene of interest. The MiMIC cassette is replaced with the FlpStop cassette by φC31-mediated cassette exchange. Insertion events are identified by the loss of the MiMIC cassette's *yellow* rescue marker from the progeny of the injected embryos (see *Venken et al., 2011a* for details). PCR is used to obtain the orientation of the insertion, and stable stocks with the FlpStop insertion in the Disrupting (D) and Non-Disrupting (ND) orientations are established. (**c**) Schematic of the FlpStop logic for conditional disruption of an endogenous gene. The cassette sits dormant within an intron of the gene of interest. While it is in the non-disrupting orientation, the splice acceptor and stop signals (stop codons and transcriptional terminators) are inverted and thus ignored (and removed during RNA splicing). In the presence of Flp recombinase, the disruptive elements are inverted and then locked in place by the FLEx switch (*Schnütgen et al., 2003*). In this disrupting locked (D lock) orientation, the splice acceptor and stop signals are revealed and disrupt expression of the gene by targeting both transcription and translation. (**d**) Schematic of how tdTomato labels mutant cells. Flies bearing one copy of a null allele and one copy of the FlpStop non-disrupting allele will be heterozygous (+/−) throughout the whole animal (left). However, in cells that express Flp recombinase, the cassette will be inverted, making it disrupting (D lock), and the tdTomato will be brought into proximity to the UAS sequence that resides outside of the FLEx switch, enabling Gal4 to drive the production of tdTomato. TdTomato therefore labels the homozygous mutant cells (−/−) (right).

*Figure 1 continued on next page*

*Figure 1 continued*

The following figure supplements are available for figure 1:

**Figure supplement 1.** FlpStop construct and example experimental crossing schemes.

**Figure supplement 2.** FlpStop CRISPR-HDR construct.

## Using FlpStop to disrupt and rescue *apterous* function

To test the ability of the FlpStop cassette to produce strong, conditional loss-of-function alleles, we targeted *apterous*, a gene encoding a transcription factor required for wing development (*Cohen et al., 1992*). The developmental role of *apterous* has been extensively characterized, and its disruption produces an easily quantifiable wing phenotype, making it a good candidate for proof-of-concept experiments. To target *apterous*, we used an intronic MiMIC insertion between exons 4 and 5, upstream of approximately half of the coding region (*Figure 2a*). Two FlpStop insertions in the *apterous* locus were isolated (non-disrupting and disrupting, *Figure 2b* and *Table 1*). We next performed complementation testing to characterize the *apterous* FlpStop alleles (designated $ap^{FlpStop\ ND}$ and $ap^{FlpStop\ D}$) with an established recessive molecular null allele, $ap^{UG035}$ (*Cohen et al., 1992*). Wings were scored on an established scale from class 1 (wild-type wing) to class 5 (no wing) (*Gohl et al., 2008*). For the FlpStop approach to be conditional, the non-disrupting orientation must be inert, maintaining wild-type gene activity, while the disrupting orientation should be strongly mutagenic, abolishing gene function. When the $ap^{FlpStop\ ND}$ allele was combined with $ap^{UG035}$, flies had almost exclusively wild-type wings (99% class 1, n = 462, *Figure 2b* and *Figure 2—figure supplement 1*). This result demonstrates that $ap^{FlpStop\ ND}$ did not interfere with gene function. Conversely, when $ap^{FlpStop\ D}$ was combined with the null allele, virtually all flies had no wings (97% class 5, n = 236, *Figure 2b* and *Figure 2—figure supplement 1*). This phenotype was as severe as homozygous flies bearing two copies of the null allele, demonstrating that $ap^{FlpStop\ D}$ completely disrupted gene function. In addition, when combined with a wild-type chromosome, neither allele produced wing abnormalities, ruling out dominant negative effects (*Figure 2—figure supplement 1*). For comparison, we note that previous studies of *apterous* using the deGradFP approach produced only class 4 wing phenotypes and were unable to completely block wing development in any flies (*Caussinus et al., 2012*). Taken together, these data demonstrate that the FlpStop cassette can produce orientation-dependent, loss-of-function alleles.

Next, we tested the ability of cassette inversion to alter gene function in a temporally controlled fashion. To invert the cassette, we expressed Flp recombinase under the control of the heat-shock promoter (*hsFlp*) (*Golic and Lindquist, 1989*). Flies that were $ap^{FlpStop\ ND}/ap^{UG035}$ or $ap^{FlpStop\ D}/ap^{UG035}$ and carried the *hsFlp* transgene were heat-shocked for 1 hr at 37°C to induce recombinase expression at different time points during development, and the phenotype of adult wings was scored (*Figure 2c*). We predicted that early manipulation of *apterous* function would have strong wing phenotypes because Apterous is essential for the development of the wing imaginal disc during larval stages (*Cohen et al., 1992*). We first tested the non-disrupting allele, whose inversion should disrupt *apterous* function. Control flies that were not heat-shocked had wild-type wings (*Figure 2d*). Experimental flies heat-shocked early in development (days 1–3) exhibited severe wing phenotypes, with many flies completely lacking one or both wings (*Figure 2d*). When cassette inversion was induced later in development, flies exhibited less severe wing phenotypes, consistent with the established role of *apterous* in wing development during late larval stages (*Wilson, 1981*). Next, we performed the converse experiment in which the disrupting allele was used to rescue *apterous* function at different developmental time points (*Figure 2e*). We observed a rescue of the *apterous* wing phenotype when the $ap^{FlpStop\ D}$ cassette was inverted early; more than half of the wings from flies heat-shocked on day 1 were wild-type (*Figure 2e*). Restoration of *apterous* later in development (days 4–7) was unable to rescue the no-wings phenotype (*Figure 2e*). Interestingly, the time window during which *apterous* is necessary for wing development differs from the time window during which its rescue is sufficient to enable wing development: heat-shock on day 4 substantially altered the phenotype of $ap^{FlpStop\ ND}/ap^{UG035}$ flies (*Figure 2d*), yet had little effect on $ap^{FlpStop\ D}/ap^{UG035}$ flies

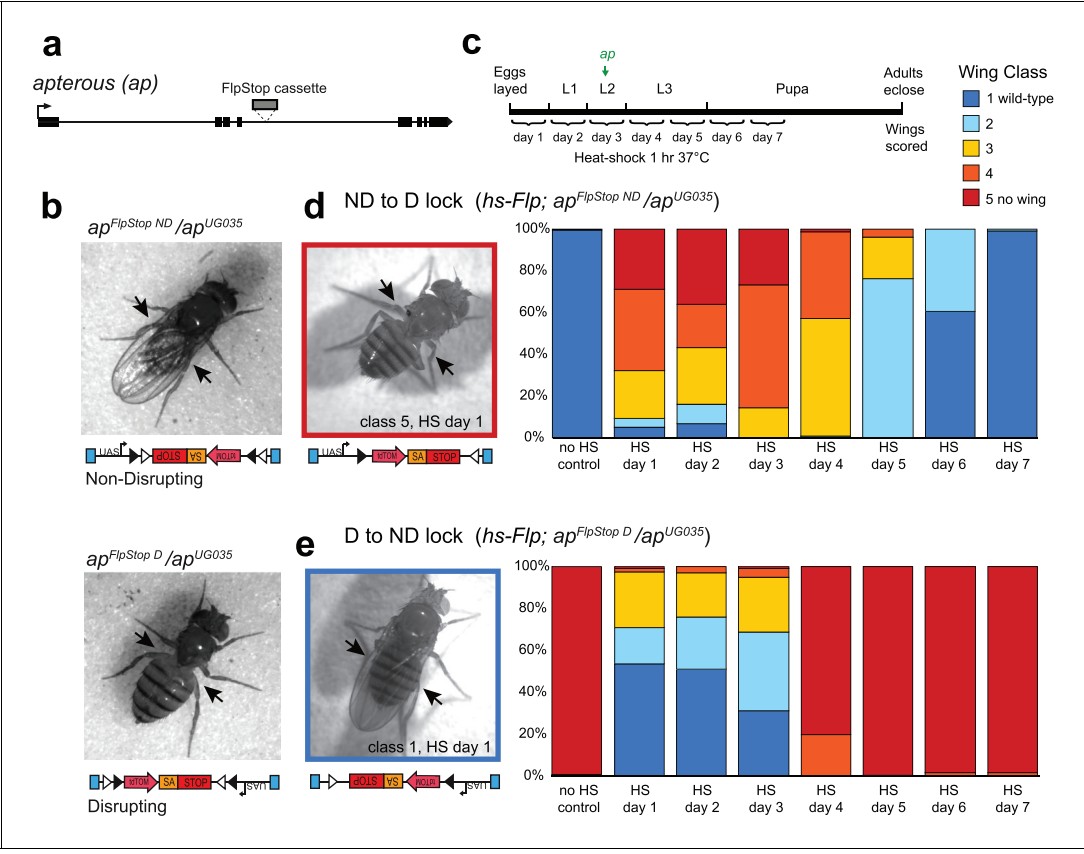

**Figure 2.** The FlpStop cassette enables both disruption and rescue of *apterous* at different developmental time points. (a) Schematic of the *apterous* locus. Black boxes denote exons, and the arrow denotes the transcriptional start site. The FlpStop cassette (gray rectangle) is inserted between exons 4 and 5. (b) Representative images of female flies that are $ap^{FlpStop\ ND}/ap^{UG035}$ (top) or $ap^{FlpStop\ D}/ap^{UG035}$ (bottom). The arrows highlight the presence or absence of wings on each fly. (c) Schematic of the experimental design. Different groups of flies bearing $ap^{FlpStop}$ alleles were heat-shocked for 1 hr at 37°C during one of the 7 days indicated in the timeline to induce Flp recombinase expression and invert the FlpStop allele. The green arrow labeled *ap* indicates that *apterous* begins to be expressed in the wing discs in mid-second instar. After eclosion, individual wings were scored on a scale ranging from class 1: wild-type wings to class 5: little or no wing tissue (**Gohl et al., 2008**). (d) Wing phenotypes of flies bearing the *apterous* non-disrupting allele ($ap^{FlpStop\ ND}$) heat-shocked at seven different time points during development (day 1-day 7) or not heat-shocked (no HS control). Full genotype: *y, w, hsFlp$^{122}$/+* or *Y; $ap^{FlpStop\ ND}/ap^{UG035}$*. n wings = 268 (no HS control), 118, 86, 112, 230, 234, 258, and 292 (day 1-day 7). An example image of an experimental fly heat-shocked on day 1 is displayed on the left. (e) Wing phenotypes of flies bearing the *apterous* disrupting allele ($ap^{FlpStop\ D}$) heat-shocked at day 1-day 7 or not heat-shocked. Full genotype: *y, w, hsFlp$^{122}$/w* or *Y; $ap^{FlpStop\ D}/ap^{UG035}$*. n wings = 154 (no HS control), 114, 108, 122, 120, 210, 176, and 204 (day 1-day 7). An example image of an experimental fly heat shocked on day 1 is displayed on the left.

The following figure supplements are available for figure 2:

**Figure supplement 1.** *apterous* complementation tests.

**Figure supplement 2.** The FlpStop cassette disrupts *apterous* cell type-specifically.

(**Figure 2e**). This difference may reflect the early requirement of *apterous* in cell fate specification; rescue of *apterous* function following misspecification of cells within the wing disc cannot restore normal wing formation (**Diaz-Benjumea and Cohen, 1993**). Taken together, these results demonstrate that FlpStop enables temporally precise gene disruption as well as gene rescue, thereby facilitating the investigation of gene function during development.

## Tissue-specific disruption of *apterous* using FlpStop

We next sought to test the efficacy of FlpStop for conditional gene disruption within a somatic mosaic clone (**Figure 2—figure supplement 2a**). Apterous protein expression during larval wing

**Table 1.** Collection of genes targeted by the FlpStop approach.

| Symbol | Gene | Function | MiMIC insertion | Location of intronic insert | Phenotype | Non-disrupting allele has phenotype? | Disrupting orientation recapitulates null phenotype*? | Substantial loss of mRNA or protein? | Tested FlpStop alleles |
|--------|------|----------|-----------------|------------------------------|-----------|--------------------------------------|-------------------------------------------------------|--------------------------------------|------------------------|
| ap | apterous | Transcription factor, wing development | MI01996 | Between exons 4 and 5 | Wingless | No | Yes – wing loss | Yes – staining | $ap^{FlpStop\ ND}$, $ap^{FlpStop\ D}$ |
| cac | cacophony | Voltage-gated Ca$^{2+}$ channel α-subunit | MI02836 | Between exons 20 and 21 | Lethal, heat paralysis with $cac^{TS2}$ | No | Yes – lethality and heat paralysis | Yes – qRT-PCR | $cac^{FlpStop\ ND}$ $cac^{FlpStop\ D\ lock}$ |
| ChAT | Choline acetyltransferase | Acetylcholine synthesis | MI04508 | Between exons 3 and 4 | Lethal | No | No | N/A | $ChAT^{FlpStop\ ND}$, $ChAT^{FlpStop\ D}$ |
| Gad1 | Glutamic acid decarboxylase 1 | GABA synthesis | MI09277 | Between exons 4 and 5 | Lethal | No | Yes – lethality | Yes – qRT-PCR | $Gad1^{FlpStop\ ND}$, $Gad1^{FlpStop\ D}$ |
| para | paralytic | Voltage-gated Na$^+$ channel α-subunit | MI08578 | Between exons 3 and 4 | Lethal, heat paralysis with $para^{ts1}$ | No | Yes – lethality and heat paralysis | Yes – qRT-PCR | $para^{FlpStop\ ND}$, $para^{FlpStop\ D}$ |
| Rdl | Resistant to dieldrin | GABA$_A$ receptor α-subunit | MI02620 | Between exons 6 and 7 | Lethal | No | Yes – lethality | Yes – qRT-PCR | $Rdl^{FlpStop\ ND}$, $Rdl^{FlpStop\ D}$ |
| Shal | Shaker cognate l | Voltage-gated K+ channel | MI00446 | Between exons 2 and 3 | None | No | N/A | Yes, 50% mRNA reduction – qRT-PCR | $Shal^{FlpStop\ ND}$, $Shal^{FlpStop\ D}$ |
| VGlut | Vesicular glutamate transporter | Glutamate packaging | MI04979 | Between exons 3 and 4 | Lethal | No | No, subviable (Hypomorph) | N/A | $VGlut^{FlpStop\ ND}$, $VGlut^{FlpStop\ D}$ |

* Measured using complementation testing.

See **Figure 3—figure supplement 2** for a description of the creation of the $cac^{FlpStop\ D\ lock}$ allele.

development is well characterized (**Figure 2—figure supplement 2b**) (**Cohen et al., 1992**; **Bieli et al., 2015a**, **2015b**). To test whether the FlpStop approach could remove Apterous protein in somatic mosaic animals, we again expressed Flp recombinase under control of the heat-shock promoter (**Golic and Lindquist, 1989**) in $ap^{FlpStop\ ND}$/$ap^{UG035}$ larvae. To mark the tissue containing the inverted cassette, the ubiquitous driver *Tubulin-Gal4* was also included. With this experimental design, *apterous* mutant cells were positively labeled with tdTomato, while the rest of the tissue was heterozygous and unmarked (**Figure 2—figure supplement 2a**). Importantly, wing discs from control larvae displayed normal Apterous expression (**Figure 2—figure supplement 2c**). TdTomato-positive regions lacked Apterous staining, while other regions of the same disc displayed high levels of protein expression (**Figure 2—figure supplement 2d**). Thus, tissue-specific inversion of the FlpStop cassette strongly impaired expression of Apterous, demonstrating the utility of the FlpStop approach for efficiently producing somatic mosaic tissue for developmental studies.

## A collection of genomic insertions for circuit interrogation

We next produced a set of FlpStop transgenic strains targeting seven genes that play central roles in either neuronal excitability or neurotransmission. All loci contained a MiMIC insertion within an intronic region common to all splice variants, upstream of a large portion of the coding sequence (**Table 1**). For neuronal excitability, we targeted *paralytic* (*para*), which encodes the alpha subunit of a voltage-gated sodium channel (**Suzuki et al., 1971**; **Loughney et al., 1989**); *cacophony (cac)*, the alpha subunit of a voltage-gated calcium channel (**Kawasaki et al., 2000**); and *Shaker cognate l* (*Shal*), a voltage-gated potassium channel (**Butler et al., 1990**). For neurotransmission, we targeted *Glutamic acid decarboxylase 1* (*Gad1*), which encodes an enzyme required for GABA synthesis (**Jackson et al., 1990**); *Resistant to dieldrin* (*Rdl*), the alpha subunit of the ionotropic GABA$_A$

receptor (*Ffrench-Constant et al., 1991*); *Vesicular glutamate transporter* (*VGlut*), a transporter that packages glutamate into synaptic vesicles (*Daniels et al., 2004*); and *Choline acetyltransferase* (*ChAT*), an enzyme that is required for acetylcholine biosynthesis (*Greenspan, 1980*). For these loci, we produced FlpStop alleles with separate insertions in both the non-disrupting and disrupting orientations (*Table 1*). Each insertion was molecularly validated to confirm successful MiMIC replacement (see Materials and methods and data not shown). For *cac*, only the non-disrupting stock was isolated by transgenesis so we instead used germline expression of Flp to isolate an allele of *cac* in which the FlpStop cassette was locked in the disrupting orientation (D lock, *Figure 3—figure supplement 2* and Materials and methods). We next extensively validated the FlpStop approach using a combination of phenotypic analysis, qRT-PCR, immunostaining, and in vivo imaging.

## FlpStop can produce conditional null alleles

To validate disruption of the targeted genes, we performed complementation testing using the non-disrupting and disrupting alleles of each gene. The genes *cac*, *ChAT*, *Gad1*, *para*, *Rdl*, and *VGlut* are essential to nervous system function; hence null alleles are homozygous lethal. We combined the disrupting and non-disrupting orientation FlpStop alleles for *ChAT*, *Gad1*, *Rdl*, and *VGlut* with known null alleles and scored adult survival (*Figure 3a,b* and *Figure 3—figure supplement 1*). Importantly, all combinations of non-disrupting alleles with null alleles were viable and consistently produced adult flies at the expected rate (*Figure 3a,b* and *Figure 3—figure supplement 1*). This demonstrated that, as required for conditional manipulations, the FlpStop construct had no phenotypic effect when in the non-disrupting orientation. In addition, we saw no evidence of dominant negative activity as measured by heterozygous lethality from the FlpStop alleles (data not shown). Next, we tested mutagenicity of the disrupting orientation. For *Gad1* and *Rdl* the combination of the disrupting allele with either of two independent null alleles was lethal, demonstrating complete disruption of gene function (*Figure 3a,b*). When placed in trans to a null allele, the *VGlut*$^{FlpStop\ D}$ allele resulted in a significant reduction in viability relative to the non-disrupting allele (*Figure 3—figure supplement 1a*). However, the combination was not completely lethal, suggesting that insertion of the FlpStop cassette created a hypomorphic allele of *VGlut* (*Figure 3—figure supplement 1a*). The *VGlut* FlpStop insertion lies between exons 3 and 4, and consistent with our observation, a chromosomal deletion of exons 1–3 creates a viable allele that only partially reduces *VGlut* function (*Daniels et al., 2006*). The FlpStop insert into a single locus, *ChAT*, was viable in combination with established null alleles, showing no evidence of gene disruption (*Figure 3—figure supplement 1b*).

*Para* and *cac* are located on the X chromosome, with null alleles being hemizygous lethal in males and homozygous lethal in females (*Ganetzky, 1984*; *Kawasaki et al., 2000*). Thus, direct complementation tests using null alleles of these genes were not feasible. FlpStop cassette insertions in the disrupting orientation in these two genes were also hemizygous and homozygous lethal, consistent with them being null alleles, while insertions in the non-disrupting orientation were viable (data not shown). Temperature-sensitive alleles of *para* and *cac* exist that display prominent paralysis upon heating (*Kawasaki et al., 2000*; *Suzuki et al., 1971*). To further test the function of the FlpStop alleles of *para* and *cac*, we performed complementation tests with these temperature sensitive alleles and measured heat-induced paralysis (*Figure 3c,d*). Under our culture conditions, flies homozygous for *para*$^{ts1}$ were sometimes immobilized when exposed to 33°C for 2 min and almost always paralyzed at 35°C (*Figure 3c*). When the cassette in the non-disrupting orientation was combined with the *para*$^{ts1}$ mutation (*para*$^{FlpStop\ ND}$/*para*$^{ts1}$), flies did not become paralyzed at either temperature, results that were comparable to control flies that were heterozygous for *para*$^{ts1}$ (*para*$^{ts1}$/+). Thus, the *para*$^{FlpStop\ ND}$ allele did not impair gene function. Conversely, flies in which the *para*$^{FlpStop\ D}$ allele was combined with *para*$^{ts1}$ (*para*$^{FlpStop\ D}$/*para*$^{ts1}$) exhibited a fully penetrant paralysis phenotype at both temperatures (*Figure 3c*). As this phenotype was more severe than that observed for *para*$^{ts1}$ homozygous mutant flies, we infer that *para*$^{FlpStop\ D}$ caused complete or nearly complete disruption of *para* function (*Figure 3c*). Analogous complementation tests were performed for the *cac*$^{FlpStop}$ alleles using the temperature-sensitive mutation *cac*$^{TS2}$ (*Kawasaki et al., 2000*) (*Figure 3d*). When *cac*$^{FlpStop}$ alleles were complemented against *cac*$^{TS2}$, the non-disrupting allele displayed little paralysis at either of two restrictive temperatures. Conversely, flies carrying the cassette in the disrupting orientation displayed a highly penetrant paralysis phenotype that was more severe than that observed in *cac*$^{TS2}$ homozygous animals. Thus, the *cac*$^{FlpStop\ D\ lock}$ allele also severely impaired gene function.

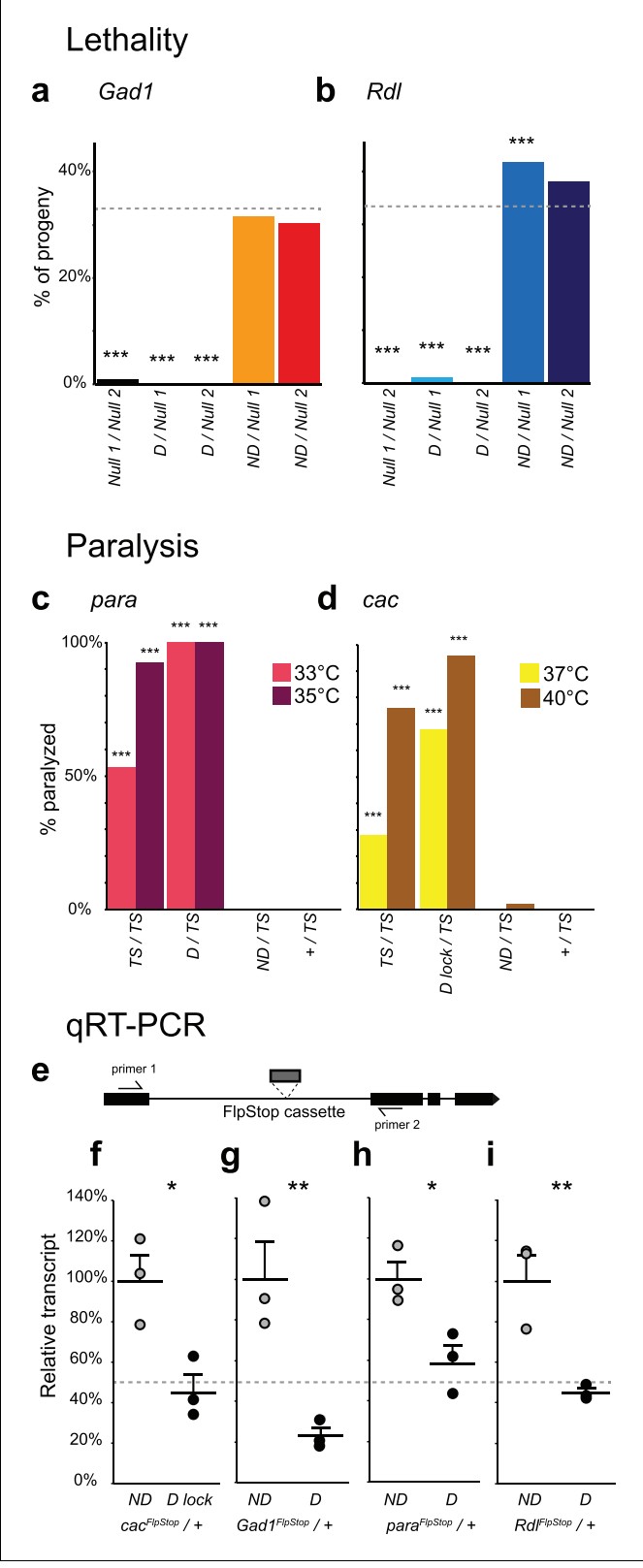

**Figure 3.** Complementation tests and qRT-PCR validate FlpStop conditional gene disruption. (**a** and **b**) Genetic interactions of FlpStop alleles with null alleles for *Gad1* (**a**) and *Rdl* (**b**) were assessed by lethality. In (**a**), *Null 1* is *Gad1*[L352F], and *Null 2* is *Df(3L)ED4341*. n flies = 251, 281, 445, 345, and 416. In (**b**), *Null 1* is *Rdl*[1], and *Null 2* is *Df (3L)Rdl-2*. n flies = 400, 267, 124, 398, and 280. FlpStop alleles are abbreviated as D and ND. Dotted line indicates

*Figure 3 continued on next page*

*Figure 3 continued*

the predicted survival of 33%. Both female and male flies were scored. Significance was assessed using a one-proportion z-test against the predicted survival. (**c** and **d**) *para* (**c**) and *cac* (**d**) FlpStop alleles were combined with temperature-sensitive mutations in these genes, and female flies were tested for paralysis upon heating. In (**c**), the *TS* allele is *para^ts1^*. n flies = 40, 14, 35, and 40 tested at 35°C, and n = 30, 14, 30, and 30 tested at 33°C. In (**d**), the *TS* allele is *cac^TS2^*. D lock is the germline-inverted *cac* FlpStop allele in the disrupting orientation. n flies = 50 per genotype and temperature. Significance was assessed using a two-tailed Fisher's exact test to compare each genotype to the *TS/+* control. (**e**) Schematic of the primer design used to assess transcript knock down using qRT-PCR. Primers were designed to amplify a ~100 bp fragment of the cDNA that flanked the insertion location of the FlpStop cassette for each gene of interest. (**f–i**) Relative transcript levels of each gene of interest from heterozygous flies bearing the non-disrupting or disrupting orientation cassette. (**f**) *cac*, (**g**) *Gad1*, (**h**) *para*, and (**i**) *Rdl*. Transcript levels were normalized to the mean of the ND sample. Means + 1 SEM as well as the individual sample measurements are plotted. 50% is denoted by the gray dotted line. An unpaired two-tailed Student's t-test was applied to the raw delta Ct values to assess significance. Heterozygous animals containing the balancer chromosome, either *FM7c* (**f** and **h**) or *TM3* (**g** and **i**), were used for both non-disrupting and disrupting conditions. *p<0.05, **p<0.01, ***p<0.001, no mark indicates p>0.05. See ***Figure 3—figure supplement 1*** for additional complementation tests and qRT-PCR, ***Figure 3—figure supplement 2*** for information about the creation of the *cac^FlpStop D lock^* allele, and ***Figure 3—source data 1*** for exact p-values.

The following source data and figure supplements are available for figure 3:

**Source data 1.** Table of statistical tests and exact p-values.

**Figure supplement 1.** Complementation tests and qRT-PCR to test FlpStop conditional gene disruption.

**Figure supplement 2.** Stable germline inversion of the *cac^FlpStop ND^* allele using *ovo-Flp*.

Next, to test whether FlpStop alleles affect mRNA transcript levels as designed, we extracted RNA from adult brains and used quantitative RT-PCR (qRT-PCR) to measure expression of each target gene. To capture the combined mutagenic effects of both the transcriptional terminators and the splice acceptor in the cassette, qRT-PCR primers were designed to amplify mRNA transcripts from the exons that flanked the FlpStop insertion (*Figure 3e*). Thus, either termination of transcription or hijacking of the splicing pattern would reduce the level of the qRT-PCR product. Complete loss of function of *Shal* is homozygous viable and has no easily scorable phenotype (*Bergquist et al., 2010*); thus, we performed qRT-PCR on flies homozygous for the FlpStop insertions to test their ability to disrupt gene function. *Shal^FlpStop D^/ Shal^FlpStop D^* flies displayed an approximately 50% reduction in *Shal* transcript relative to *Shal^FlpStop ND^/ Shal^FlpStop ND^* flies, consistent with the *Shal^FlpStop D^* allele functioning as a hypomorph (*Figure 3—figure supplement 1c*). We also wanted to confirm that the alleles that were lethal by complementation testing (*cac*, *Gad1*, *para*, and *Rdl*) fully reduced mRNA transcript levels. As the disrupting alleles of these genes were homozygous lethal, we measured gene expression in heterozygous animals bearing one copy of either the non-disrupting or disrupting allele. Under these conditions, if the disrupting allele were behaving as a null, heterozygous *D/+* flies should express half as much intact target gene mRNA as heterozygous *ND/+* flies. Consistent with our complementation tests, transcripts in heterozygous flies bearing the disrupting allele were reduced approximately 50% relative to heterozygous flies bearing the non-disrupting allele (*Figure 3f–i*).

In summary, we created a set of FlpStop transgenic strains targeting seven genes that play central roles in neuronal communication. For all seven genes, as predicted, the non-disrupting alleles had no phenotypic effect (*Table 1*). For four of the seven genes, *cac*, *Gad1*, *para*, and *Rdl*, the disrupting orientations were genetically null (*Table 1*). For two of the seven genes, *VGlut* and *Shal,* the disrupting alleles were hypomorphic (*Table 1*). A single locus, *ChAT*, showed no evidence of gene disruption. Taken together, these genetic and molecular tests of the FlpStop alleles demonstrate that non-disrupting alleles have no negative effect on gene activity and that disrupting alleles often behave as genetic null or hypomorphic alleles (six of seven genes). Thus, the FlpStop approach can produce conditional loss-of-function mutations.

## Labeling by the fluorescent reporter indicates efficient, cell type-specific cassette inversion

In the FlpStop approach, only cells that have both undergone cassette inversion and express Gal4 protein will be labeled by tdTomato (*Figure 4a*). As a result, this design can be used to positively label mutant cells. To directly test the efficiency of cassette inversion, we expressed Flp recombinase in different neuronal populations using different Gal4 driver lines in animals bearing $cac^{FlpStop\ ND}$. Flipping efficiency was quantified by counting the number of Gal4-expressing cells, as labeled by *UAS*-driven expression of GCaMP6f, that also expressed tdTomato. We tested cassette inversion using three different driver lines that label the neurons Mi1, Tm1, and T4 in the visual system, respectively (*Jenett et al., 2012*; *Strother et al., 2014*; *Maisak et al., 2013*). When Flp recombinase was expressed in flies carrying $cac^{FlpStop\ ND}$, creating $cac^{FlpStop\ D\ lock}$ via cell type-specific inversion, tdTomato expression was observed throughout the cell bodies and neurites, labeling 90%, 93%, and 70% of GCaMP6f-expressing cells (*Figure 4b–d*). Cells in which tdTomato immunolabeling was discernable but GCaMP6f signal was absent were extremely rare in all genotypes tested (<0.01%, data not shown).

We next tested the inversion efficiency of each of the eight FlpStop alleles by using the Mi1 Gal4 line to express Flp recombinase and the fluorescent marker mCD8::GFP. The FlpStop cassettes in *ap*, *cac*, *ChAT*, *para*, *Rdl*, and *Shal* all inverted efficiently: depending on the gene, 76% to 96% of GFP-positive cells were also tdTomato-positive (*Figure 5*). However, we observed that there was no tdTomato expressed in Mi1 from the *Gad1* and V*Glut* FlpStop alleles (*Figure 5—figure supplement 1a,b*). As this result contrasted sharply with the efficient inversion of the other alleles, we used the pan-neuronal driver *nSyb-Gal4* to express Flp recombinase to test whether the cassettes in *Gad1* and *VGlut* were able to be inverted in other neurons. We observed tdTomato in many neurons in the visual system, indicating that these FlpStop alleles are capable of inverting and expressing the tdTomato reporter (*Figure 5—figure supplement 1c,d*). Notably, the expression patterns were distinct between the two genes even though they were consistent across brains of the same genotype (*Figure 5—figure supplement 1c,d* and data not shown). We speculate that silencing of the genetic locus when a gene is not expressed in a neuron (as the *VGlut* and *Gad1* loci might be in a cholinergic neuron such as Mi1) also silences expression of tdTomato, despite the presence of Gal4. Indeed, *UAS* transgene expression is well known to depend on the site of integration into the genome (*Markstein et al., 2008*).

Taken together, these results demonstrate that FlpStop cassette inversion is efficient across genes and can be achieved when the recombinase is expressed using a number of different Gal4 driver lines, demonstrating compatibility with existing tools for targeting and manipulating specific neuronal populations.

## *Cac* selectively alters visually evoked calcium signals in individual neuronal compartments

The ability to selectively manipulate gene activity in single identified neurons within intact circuits enables investigation of how specific genes contribute to neuronal function. To establish this proof-of-concept for FlpStop, we examined visually evoked responses in the third-order visual interneuron Tm3. This small monopolar neuron has branched arborizations in many brain layers, including M1, M5, M8, and M10 of the medulla and layers Lo1 and Lo4 of the lobula (*Figure 6a*) (*Fischbach and Dittrich, 1989*; *Hasegawa et al., 2011*). The layer M1 and M5 arbors are primarily dendritic, receiving the majority of their inputs from the neuron L1, which in turn receives direct input from the R1-6 photoreceptors (*Takemura et al., 2013*; *Meinertzhagen et al., 1991*). The arbors in layers M8, M10, Lo1, and Lo4 are primarily presynaptic, with prominent synapses onto T4 in layer M10 (*Takemura et al., 2013*; *Hasegawa et al., 2011*). Although each Tm3 arbor is primarily either dendritic or axonal, each arbor can contain both pre- and postsynaptic sites (*Takemura et al., 2013*). Tm3 responds with transient graded depolarizations to increases in light intensity (ON) and is required for flies to respond behaviorally to fast moving ON edges (*Behnia et al., 2014*; *Yang et al., 2016*; *Ammer et al., 2015*). This depolarization is accompanied by a corresponding increase in calcium concentration, but interestingly, unlike the voltage signals, these calcium signals vary non-uniformly across the neuron's arbors, indicating that they are compartmentalized (*Yang et al., 2016*).

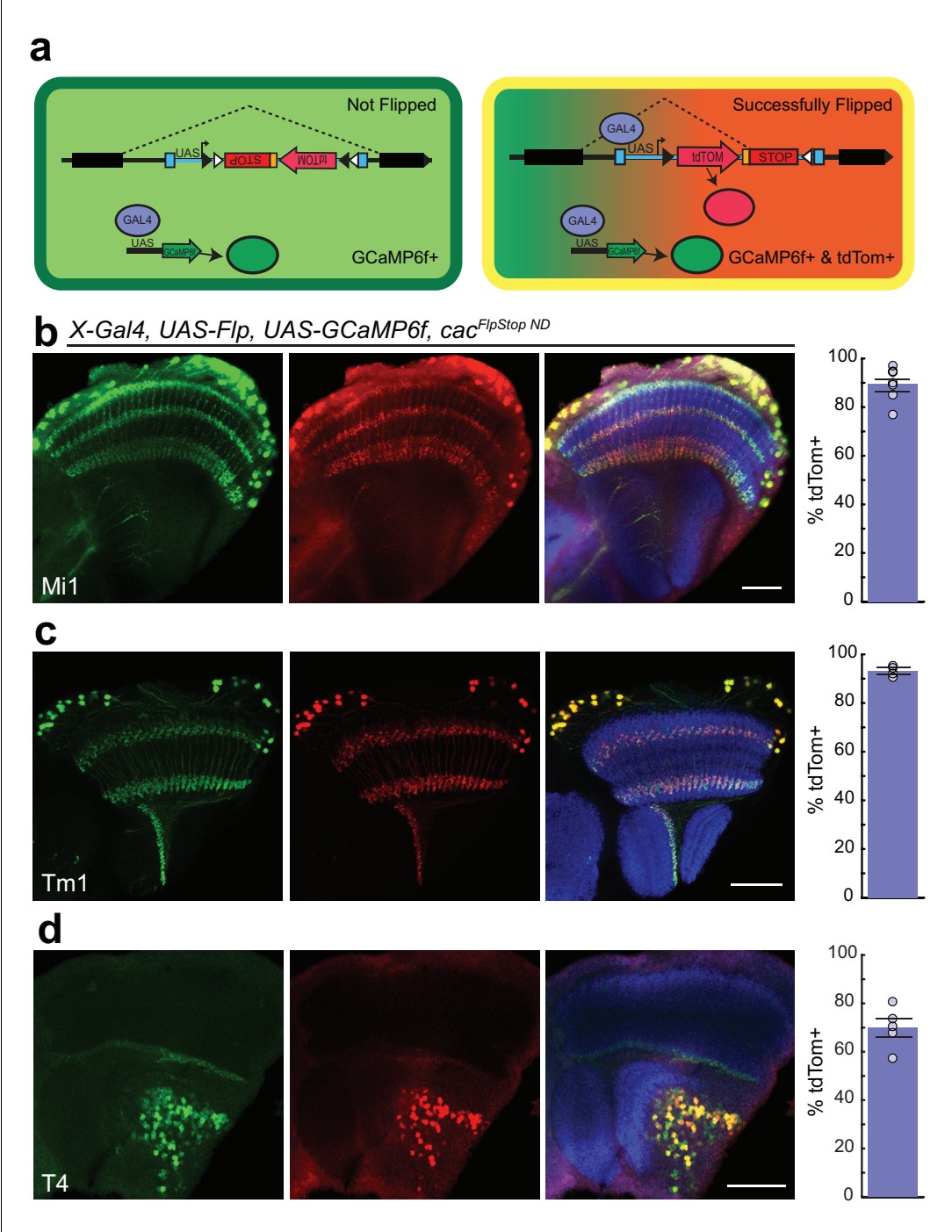

**Figure 4.** Cassette inversion is efficient across Gal4 driver lines. (a) Schematic of the experimental logic used to test the inversion of the FlpStop cassette. The full expression pattern of the Gal4 driver line is labeled by GCaMP6f (green) while tdTomato and GCaMP6f together (yellow) label the subset of the Gal4 pattern in which Flp recombinase has inverted the cassette and has 'Successfully Flipped'. (b–d) Confocal images of adult brains with GCaMP6f (anti-GFP, green, left), and tdTomato (anti-dsRed, red, middle), or a merge including Bruchpilot (nc82, blue, right). A bar plot displaying the percentage of GCaMP6f-positive neurons that are also labeled by tdTomato is shown on the far right. The bar is the mean, the error bars are ±1 SEM, and the dots are the values for each individually scored visual system. (b) Visual system Mi1 neurons labeled with $Mi1^{R19F01}$-Gal4. Full genotype: $cac^{FlpStopND}$/+; UAS-Flp/UAS-GCaMP6f; $Mi1^{R19F01}$-Gal4/+. N visual systems = 7, N cells scored = 775. (c) Visual system Tm1 neurons labeled with $Tm1^{R74G01}$-Gal4. Full genotype: $cac^{FlpStopND}$/w; UAS-Flp/UAS-GCaMP6f; $Tm1^{R74G01}$-Gal4/+. N visual systems = 4, N cells scored = 576. (d) Visual system T4 neurons labeled with $T4^{R54A03}$-Gal4. Full genotype: $cac^{FlpStopND}$/+; UAS-Flp/UAS-GCaMP6f; $T4^{R54A03}$-Gal4/+. N visual systems = 5, N cells scored = 494. All images are maximum intensity projections of ~5–10 μm z-stacks. Scale bars are 30 μm.

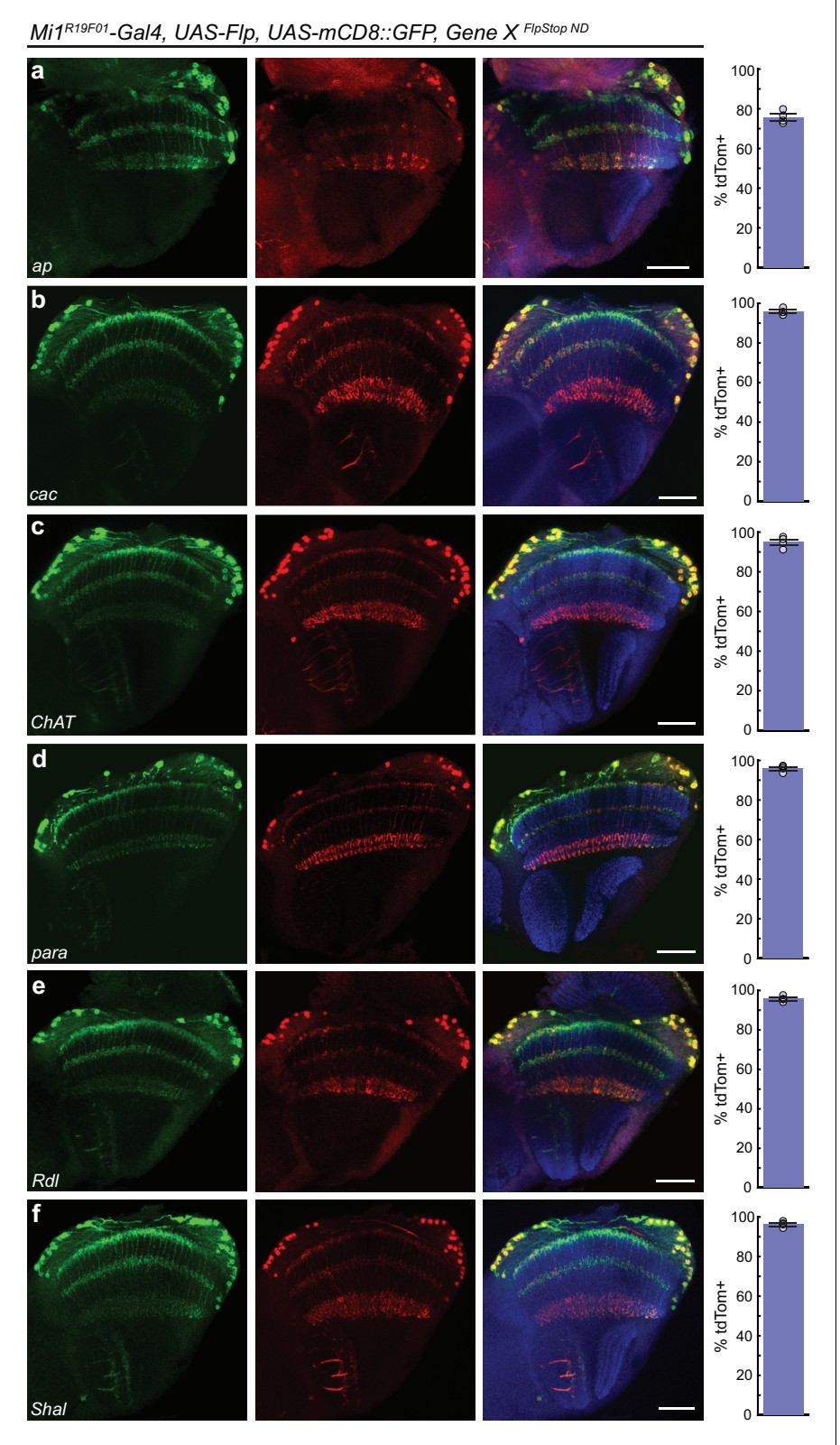

**Figure 5.** FlpStop alleles of each gene invert and express tdTomato. (**a–f**) Confocal images of adult brains stained for mCD8::GFP (anti-GFP, green, left), tdTomato (anti-dsRed, red, middle), and a merge including Bruchpilot (nc82, blue, right). *Mi1*$^{R19F01}$-*Gal4* was used to drive *UAS-Flp* and *UAS-mCD8::GFP* and was combined with the non-disrupting FlpStop alleles of (**a**) *ap*, (**b**) *cac*, (**c**) *ChAT*, (**d**) *para*, (**e**) *Rdl*, and (**f**) *Shal*. All images are maximum

*Figure 5 continued on next page*

*Figure 5 continued*

intensity projections of ~5–10 μm z-stacks. Scale bars are 30 μm. To the right of each example confocal image, bar plots display the percentage of mCD8::GFP positive neurons that are also labeled by tdTomato. The bar is the mean, the error bars are ±1 SEM, and the dots are the values for each individually scored visual system. Sample numbers denoted as N visual systems (N cells scored) are (a) *ap*: 4 (906), (b) *cac*: 4 (714), (c) *ChAT*: 4 (992), (d) *para*: 4 (676), (e) *Rdl*: 4 (676), (f) *Shal*: 4 (1025).

The following figure supplement is available for figure 5:

**Figure supplement 1.** FlpStop alleles of *Gad1* and *VGlut* invert and express tdTomato.

---

We hypothesized that voltage-gated calcium channels contribute to Tm3's light-evoked calcium signals and that differential distribution or regulation of specific types of calcium channels underlies compartmentalization. Given Tm3's complex morphology and striking pattern of calcium signals, we tested the role of the voltage-gated calcium channel *cac* in producing its visually evoked calcium responses. To do this, we performed two-photon imaging of the genetically encoded calcium indicator GCaMP6f expressed in Tm3 and measured impulse responses to brief flashes of light.

We first examined whether the $cac^{FlpStop\ ND}$ allele was inverted efficiently by the Tm3 driver line. As our Tm3 driver is expressed in other cells during larval stages (*Jenett et al., 2012*), removal of *cac* under these conditions was developmentally lethal. Thus, we incorporated a $tub\text{-}Gal80^{ts}$ transgene, a ubiquitously expressed temperature-sensitive repressor of Gal4 (*McGuire et al., 2003*), to restrict Gal4 activity to late pupal and adult stages. We compared experimental males hemizygous for $cac^{FlpStop\ ND}$ and carrying *UAS-Flp*, *UAS-GCaMP6f*, $tub\text{-}Gal80^{ts}$, and *Tm3-Gal4* to two separate controls: males lacking *UAS-Flp* (no-Flp control) and heterozygous females carrying one copy of $cac^{FlpStop\ ND}$ and one wild-type *cac* allele (heterozygous control). The no-Flp control should neither disrupt *cac* nor express tdTomato but otherwise contains all other transgenes, thereby controlling for their presence. The heterozygous control should invert $cac^{FlpStop\ ND}$ and express tdTomato; however, as it carries one wild-type *cac* allele, *cac* function in Tm3 likely remains largely intact. This controls for any potential effects of Flp and tdTomato on Tm3. Experimental males and heterozygous controls expressed tdTomato in nearly all Tm3 cells while no-Flp controls expressed no detectable tdTomato (*Figure 6b–e* and data not shown). In one-day old adults, nearly all of Tm3 neurons expressed tdTomato and therefore carried the inverted FlpStop cassette (data not shown). To provide sufficient time for the FlpStop manipulation to overcome possible protein perdurance given estimated ion channel half-lives (*Passafaro et al., 1992*), we performed all subsequent experiments in 10–12 day old adult flies.

In larval motor neurons, loss of *cac* function substantially reduces axon branching and synaptic bouton number (*Rieckhof et al., 2003*). We therefore examined whether FlpStop-induced loss of Cac altered the morphology of Tm3. However, confocal imaging revealed no structural changes (*Figure 6f–k*). Thus, any observed changes in calcium signals following *cac* disruption are not the result of gross anatomical disruptions in Tm3 morphology. We also quantified cassette inversion efficiency in these flies: 91% of cells were both tdTomato- and GCaMP6f-positive (*Figure 6l*). We restricted all calcium imaging analysis to Tm3 arbors that were positively labeled by tdTomato, a signal that was easily visible during in vivo imaging (*Figure 6b–e*).

To measure visual stimulus-evoked Tm3 calcium signals, we presented 25 ms light flashes off of a gray background and used two-photon microscopy to image changes in GCaMP6f fluorescence in each of the Tm3 arbors as well as the cell body. Across the cell, Tm3 responded to this stimulus with a transient increase in calcium concentration (*Figure 7*) (*Yang et al., 2016*). Compared with both no-Flp and heterozygous controls, Tm3 cells lacking Cac displayed significantly reduced responses in the cell body as well as in the arbors in layers Lo1 and Lo4 (*Figure 7a,f,g*). This suggests that in these cellular compartments, Cac is required for a substantial portion, but not all, of the visually evoked calcium response. Strikingly, calcium signals in the layer M5 arbor became dramatically larger when Cac was removed (*Figure 7c*). We speculate that removal of Cac from this compartment leads to replacement of Cac channels with one or more other channel types, resulting in increased calcium influx. Interestingly, loss of *cac* function did not affect responses in the arbors of layers M1, M8, and M10 (*Figure 7b,d,e*), demonstrating that calcium signals in these compartments can be

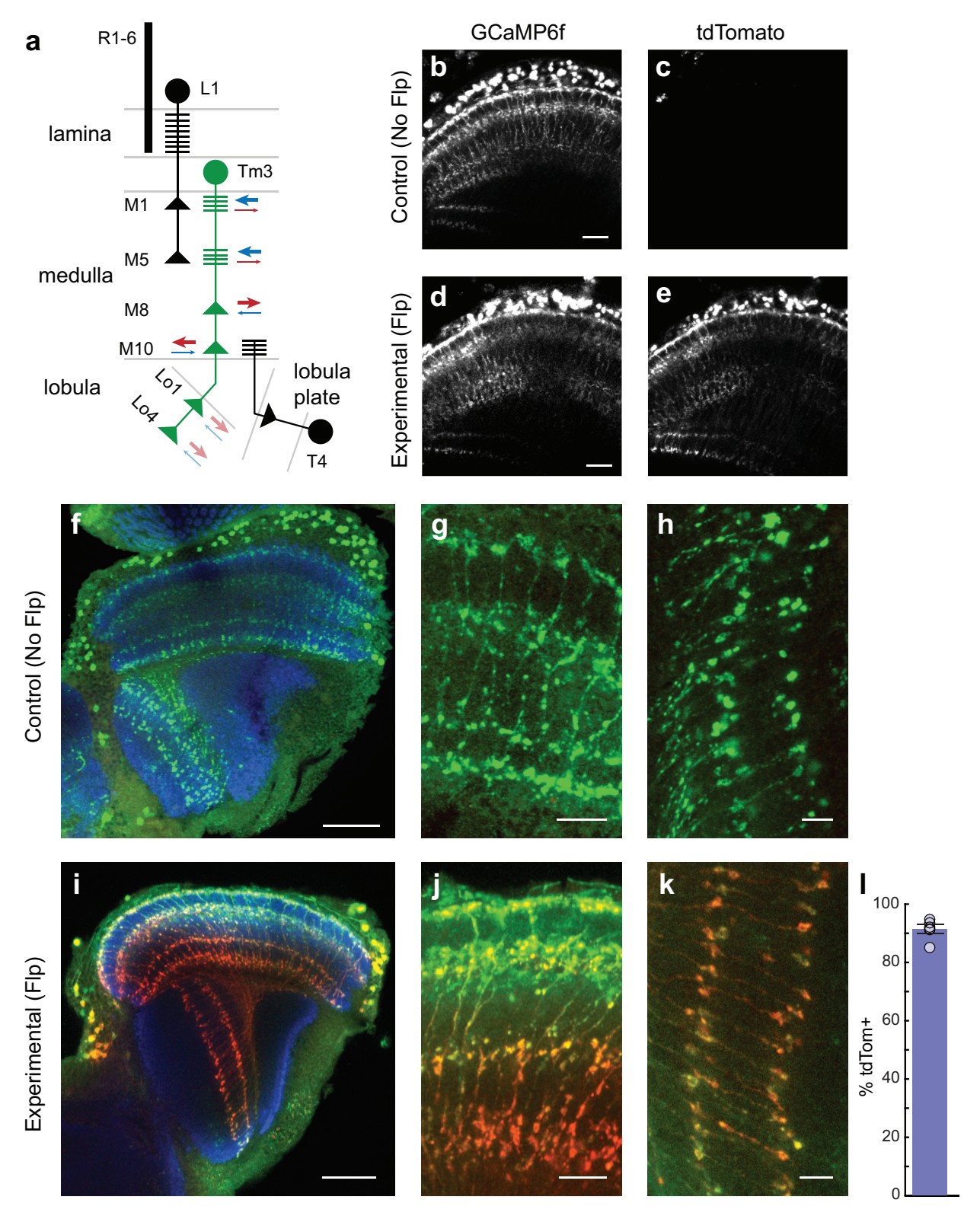

**Figure 6.** *cac^{FlpStop ND}* inverts efficiently in Tm3 neurons, and cell type-specific loss of Cac does not disrupt cell morphology. (a) Schematic of the transmedulla visual neuron Tm3 (green) and selected local circuitry. In the lamina neuropil, R1-6 photoreceptors synapse onto L1, which in turn provides input onto Tm3 in medulla layers M1 and M5. Tm3 synapses onto the direction-selective neuron T4 in layer M10. The arrows depict the relative fraction of input (blue) and output (red) synapses at each Tm3 arbor, with heavier weights indicating a larger contribution. Darker arrows represent connections

*Figure 6 continued on next page*

*Figure 6 continued*

identified by EM reconstruction (*Takemura et al., 2013*). Lighter arrows are hypothesized synaptic contributions based on Syt-HA labeling (*Hasegawa et al., 2011*). (**b–e**) In vivo expression of GCaMP6f (**b** and **d**) and tdTomato (**c** and **e**). In (**b** and **c**) a control lacking Flp (no-Flp control) is shown (full genotype: *cac*$^{FlpStop\ ND}$/Y; +/UAS-GCaMP6f; Tm3 $^{R13E12}$-Gal4/tubP-Gal80$^{ts}$), and in (**d** and **e**) an experimental animal is shown (full genotype: *cac*$^{FlpStop\ ND}$/Y; UAS-Flp/UAS-GCaMP6f; Tm3 $^{R13E12}$-Gal4/tubP-Gal80$^{ts}$). One-photon excitation was used. Images are maximum intensity projections of 6 µm z-stacks. Scale bars are 20 µm. (**f–k**) Confocal images of the visual system from no-Flp control (**f–h**) and experimental (**i–k**) flies. GCaMP6f is labeled in green (anti-GFP), tdTomato is labeled in red (anti-dsRed), and the neuropil is labeled in blue (anti-Bruchpilot). (**f** and **i**) Full optic lobes. (**g** and **j**) Tm3 arbors in medulla layers M1, M5, M8, and M10. (**h** and **k**) Arbors in lobula layers Lo1 and Lo4. Images are maximum intensity projections of ~5–10 µm z-stacks. Scale bars are 30 µm in (**f** and **i**), 10 µm in (**g** and **j**), and 5 µm in (**h** and **k**). (**l**) Bar plot displays the percentage of GCaMP6fpositive Tm3 neurons that are also labeled by tdTomato. The bar is the mean, the error bars are ±1 SEM, and the dots are the values for each individually scored visual system. N visual systems = 6, N cells scored = 467.

mediated by other channel types. Taken together, these results demonstrate that Cac is a critical regulator of Tm3's calcium responses to visual stimuli and has differential effects across cellular compartments.

## Discussion

Cell type-specific gene manipulation enables a wealth of experiments addressing critical questions in many different contexts. Here we describe FlpStop, a generalizable strategy for producing cell type-specific disruption or rescue of gene function that is reported with fluorescent protein expression. We demonstrate that insertion of the FlpStop cassette into coding introns using the MiMIC collection enables straightforward production of conditional null alleles of many genes. Conditional gene disruption with FlpStop can be easily targeted to cell populations of interest through expression of Flp recombinase using the Gal4 system. The fluorescent reporter provides a readout of cassette orientation and is compatible with in vivo and fixed-tissue imaging. To make a broadly useful toolkit for dissecting neural circuit function, we created and validated FlpStop alleles of genes involved in GABAergic inhibition (*Rdl* and *Gad1*) as well as voltage-gated conductances (*para* and *cac*). In a proof-of-principle experiment, we used this tool to disrupt *cac* within Tm3 neurons. Strikingly, this experiment uncovered a novel role for *cac* in select subcellular compartments, demonstrating not only the complexity of calcium signaling but also the utility of the FlpStop approach. Thus, FlpStop is a powerful tool for targeted gene manipulation to address questions exploring the interactions between genes, circuits, and behavior.

### The current FlpStop tool kit

We generated FlpStop alleles of eight genes and extensively tested their ability to disrupt gene function. Of these eight, six were successful. The disrupting alleles of *ap*, *cac*, *Gad1*, *para*, and *Rdl* were molecular nulls, and the disrupting allele of *VGlut* was a hypomorph, as predicted from previous analysis of mutations in the *VGlut* locus (*Daniels et al., 2006*). Two genes targeted by FlpStop were not completely successful in creating the desired conditional null alleles: the insert into *Shal* produced a hypomorph, and the insert into *ChAT* showed no evidence of gene disruption. Interestingly, *ChAT* is a complex genetic locus where the coding region of *VAChT* is nested within the first intron of the *ChAT* gene, and it has been proposed that both transcripts arise from a common transcript by differential RNA processing (*Kitamoto et al., 1998*). Perhaps this complex, locus-specific regulation makes it insensitive to the mutagenic signals embedded within the FlpStop cassette.

Furthermore, we tested that FlpStop alleles in the non-disrupting orientation were indeed inert, a critical feature that enables conditional manipulation of the genes. For all eight genes, we found that the non-disrupting alleles did not interfere with gene function. Finally, we also demonstrated that these FlpStop alleles all inverted efficiently in the presence of Flp recombinase and expressed tdTomato. Taken together, our data suggest that incorporation of the FlpStop cassette will produce a successful conditional allele for many genes of interest.

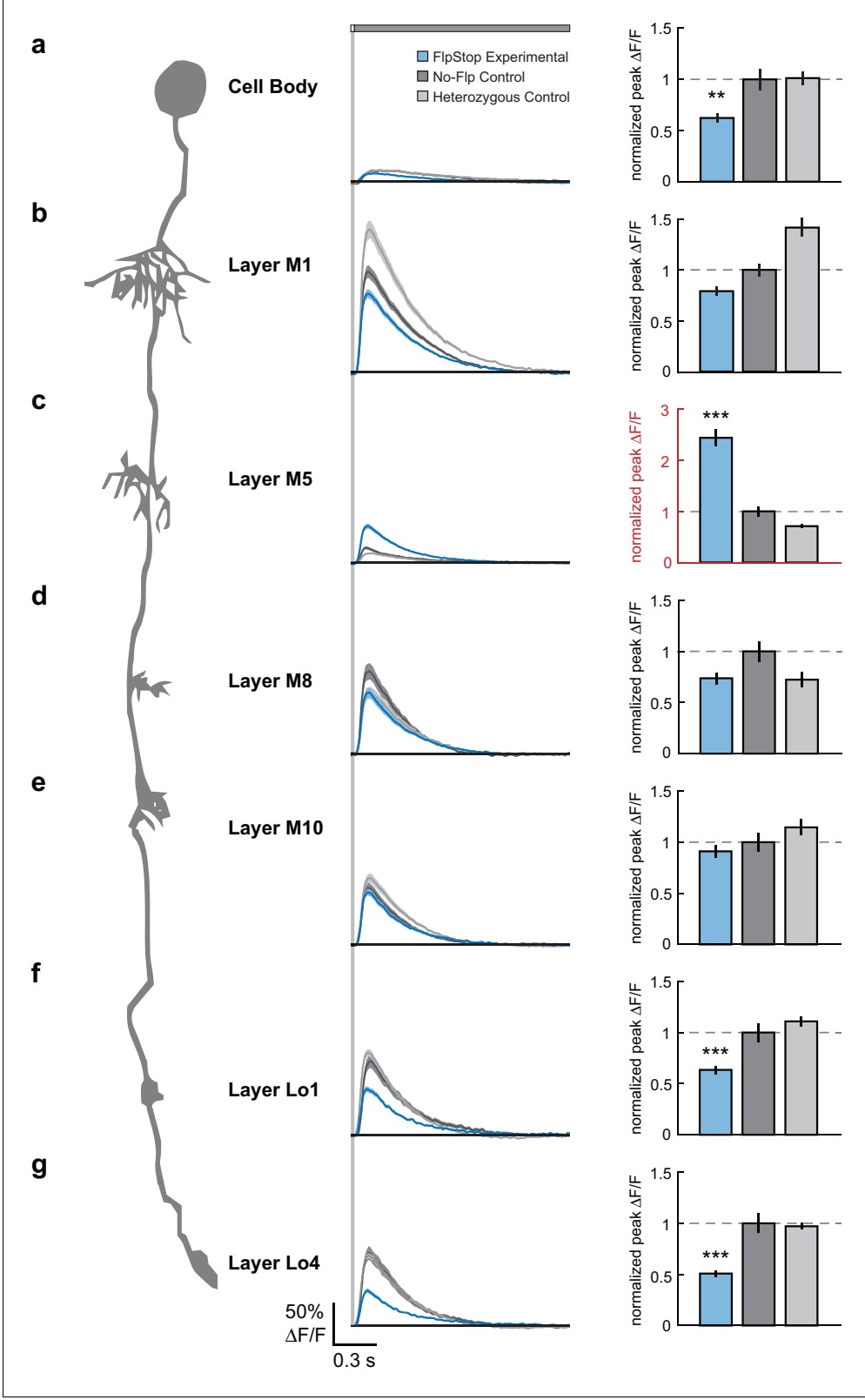

**Figure 7.** Loss of Cac from Tm3 alters visually evoked calcium signals in a compartment-specific manner. (a–g) Left: Schematic of Tm3. Middle: calcium responses in Tm3 from experimental (blue), no-Flp control (dark gray), and heterozygous control (light gray) flies to a 25 ms light flash with a 1500 ms gray interleave, contrast = 0.5. The solid line is the mean response; the shaded region is ±1 SEM. Right: Peak ΔF/F normalized to the mean of the no-

*Figure 7 continued on next page*

*Figure 7 continued*

Flp control. The mean ±1 SEM is plotted. Note in red the different y-axis scale in (c). **p<0.01, ***p<0.001 (Student's two-sample t-test, Bonferroni correction for multiple comparisons). Asterisks are displayed only if the experimental condition is significantly different from both controls. If the two comparisons had different p-values, the less significant one is shown. The imaged regions were: (a) cell body (experimental: n = 26 cells, 6 flies; no-Flp control: n = 22 cells, 5 flies; heterozygous control: n = 38 cells, 6 flies); (b) layer M1 arbor (experimental: n = 58 cells, 6 flies; no-Flp control: n = 56 cells, 6 flies; heterozygous control: n = 67 cells, 6 flies); (c) layer M5 arbor (experimental: n = 75 cells, 3 flies; no-Flp control: n = 62 cells, 3 flies; heterozygous control: n = 59 cells, 3 flies), (d) layer M8 arbor (experimental: n = 63 cells, 7 flies; no-Flp control: n = 42 cells, 6 flies; heterozygous control: n = 45 cells, 7 flies), (e) layer M10 arbor (experimental: n = 71 cells, 9 flies; no-Flp control: n = 81 cells, 9 flies; heterozygous control: n = 78 cells, 9 flies), (f) layer Lo1 arbor (experimental: n = 49 cells, 7 flies; no-Flp control: n = 41 cells, 6 flies; heterozygous control: n = 67 cells, 7 flies), and (g) layer Lo4 arbor (experimental: n = 44 cells, 5 flies; no-Flp control: n = 39 cells, 6 flies; heterozygous control: n = 67 cells, 7 flies). 9 flies were imaged for each genotype; multiple regions were imaged in each fly. The full genotypes are: experimental: $cac^{FlpStop\ ND}$/Y; UAS-Flp/UAS-GCaMP6f; $Tm3^{R13E12}$-Gal4/tubP-Gal80$^{ts}$; no-Flp control: $cac^{FlpStop\ ND}$/Y; +/UAS-GCaMP6f; $Tm3^{R13E12}$-Gal4/tubP-Gal80$^{ts}$; and heterozygous control: $cac^{FlpStop\ ND}$/ w; UAS-Flp/UAS-GCaMP6f; $Tm3^{R13E12}$-Gal4/tubP-Gal80$^{ts}$.

The following source data is available for figure 7:

**Source data 1.** Table of exact p-values.

## Targeting additional genetic loci using FlpStop

FlpStop can be readily generalized to many genes of interest. At present, MiMIC insertions allow access to the coding introns of approximately 24% of neuronal genes (as annotated by flybase.org; see Materials and methods) (*Nagarkar-Jaiswal et al., 2015b*). For genes that lack a suitable MiMIC site, the FlpStop construct can be targeted to desired genomic loci using CRISPR/Cas9-mediated homology-directed repair (CRISPR-HDR). To facilitate this approach, we have created a plasmid for integration of the FlpStop construct through CRISPR-HDR (*Cong et al., 2013*; *Gratz et al., 2014*; *Mali et al., 2013*) (Materials and methods, *Figure 1—figure supplement 2*). This plasmid contains a multiple cloning site on either side of the FlpStop cassette to enable inclusion of the homology arms and a Cre recombinase-excisable *3xP3-dsRed* reporter to enable scoring of successful incorporation of the cassette into the genome (*Gratz et al., 2014*) (*Figure 1—figure supplement 2*). Given the flexible access to almost all genetic loci provided by CRISPR-HDR, the FlpStop cassette can be targeted to essentially any gene of interest that contains a coding intron. While more time consuming because of the need to clone new homology arms for each genetic locus, CRISPR-HDR-mediated insertion of FlpStop is more flexible than MiMIC-based insertion and could be used to target specific introns for more efficient gene disruption. The CRISPR-HDR approach can be further adapted to target genomic regions containing a Gal4 transgene using Homology Assisted CRISPR Knock-in, opening up additional loci without the need for gene-specific homology arms (*Gratz et al., 2014*; *Lin and Potter, 2016*). As the FlpStop tool kit is expanded to more loci, more information about the failures and successes of insertions will be obtained. Incorporating this new information into the choice of intron for the targeting of new genes could further increase the already high success rate of the FlpStop approach.

## FlpStop enables a diversity of applications

Notably, the FlpStop cassette can initially be integrated into the gene in either the non-disrupting or the disrupting orientation such that Flp-mediated inversion then creates locked disrupting and non-disrupting alleles, respectively. Each of these inverted alleles would express tdTomato under Gal4 control. In this work, we demonstrate that the transformation from non-disrupting to disrupting provides a powerful approach for exploring the functions of essential genes. For non-essential genes, the transformation from disrupting to non-disrupting allows for cell type-specific genetic rescue by the endogenous locus, as demonstrated by the rescue of *apterous* function at specific time points during development (*Figure 2*). For this application, one should combine the disrupting allele with a null allele so that the entire animal lacks gene activity but the Flp-targeted cell population becomes

heterozygous. This approach offers advantages over rescue experiments using the Gal4 system to express cDNAs as the genes are expressed in their native regulatory contexts. Therefore, it will be particularly useful for studies of gene function during development and circuit studies of genes involved in, for example, sensory transduction.

Expression of Flp recombinase provides the FlpStop approach both spatial and temporal control. For example, as we demonstrate with $ap^{FlpStop}$, use of the heat-shock promoter to express Flp restricts inversion of the FlpStop construct to a brief time window, a manipulation that is well suited to developmental studies (*Figure 2* and *Figure 2—figure supplement 2*). Flp recombinase can also be expressed under Gal4 control, and additional temporal specificity can be achieved by incorporating Gal80$^{ts}$, as we demonstrate with Tm3 (*Figures 6* and *7*). These approaches enable the inversion of the FlpStop construct to be restricted to a time period of hours to days. In addition, combining the FlpStop approach with temperature-sensitive alleles enables control of gene activity on the scale of seconds to minutes (*Siddiqi and Benzer, 1976*; *Suzuki et al., 1971*). As we demonstrated with *para* and *cac* (*Figure 3*), one can place a temperature-sensitive allele in trans to the FlpStop allele of the same gene and use a temperature shift to disrupt gene function. Extending this approach, when the disrupting orientation is induced within a cell type of interest, those cells will only produce proteins containing the temperature-sensitive mutation while all other cells will express both wild-type and mutant protein. Upon temperature shift, the cell type of interest will lack functional protein while the rest of the animal will have a 50% reduction in gene activity, becoming effectively heterozygous. This strategy could be applied to any gene where a temperature-sensitive mutation exists, thereby enabling highly precise spatial and temporal control of gene activity.

FlpStop complements and extends existing techniques for cell type-specific gene manipulation. First, our validated FlpStop disrupting alleles are nulls, and the non-disrupting alleles have wild-type function, reducing the experimental variability that can emerge when performing gene knock down using RNAi or deGradFP. That is, in these two approaches, differences in the strength of the Gal4 driver result in differences in the extent of gene knock down in each cell. In contrast, because of the binary nature of FlpStop, differences in the strength of the driver change the frequency of cassette inversion, producing different proportions of mutant cells; the extent of gene disruption across mutant cells is constant. However, RNAi requires fewer genetic elements than FlpStop, and collections targeting most genes are available (*Dietzl et al., 2007*). RNAi therefore may be more suitable for large-scale screens and experiments that contain genetic complexity in addition to gene knock down, while FlpStop would be more suitable for targeted interrogations where complete removal or rescue of gene function is required. Second, unlike recombinase-mediated deletion of exons, RNAi, deGradFP, and CRISPR/Cas9, FlpStop positively labels affected cells with a fluorescent reporter of gene manipulation and enables gene rescue in addition to gene disruption. Third, unlike MARCM, which also positively labels manipulated cells, FlpStop can be used to target all the cells of a single differentiated type in the adult brain. However, like MARCM, FlpStop requires a null allele of the gene of interest, though for FlpStop, this issue can be avoided for genes on the X chromosome by the use of hemizygous males (*Figures 6* and *7*). For autosomal genes, a rich history of *Drosophila* genetics has produced null alleles of many genes. When choosing a null for FlpStop analysis of an autosomal gene, we recommend avoiding alleles that contain an FRT site, as these could recombine with the FlpStop cassette and generate unwanted DNA rearrangements. This includes the disrupting FlpStop allele of the same gene and some deficiencies. If an established null does not already exist for a gene of interest, MiMIC insertions in the disrupting orientation can often provide a null allele, and CRISPR-Cas9 technology enables the straightforward production of null alleles by creating targeted mutations within an essential exon (*Venken et al., 2011a*; *Bassett et al., 2013*). Thus, FlpStop opens a number of new experimental avenues.

## Cac differentially regulates calcium signals across a neuron

To demonstrate the utility of the FlpStop approach to studying neuronal function in vivo, we removed Cac specifically from the visual interneuron Tm3. We found that Cac is a critical regulator of light-evoked calcium responses in specific subcellular compartments (*Figure 7*). In motor neurons, Cac is the voltage-gated calcium channel required presynaptically for evoked neurotransmitter release (*Kawasaki et al., 2000*, *2002*; *Kawasaki and Zou, 2004*). However, recent studies in olfactory projection neurons (PNs) demonstrate that Cac may have a more complex role in the central nervous system (*Gu et al., 2009*). We measured visually evoked calcium signals in all of the arbors of

Tm3, comprising four predominately axonal regions and two predominately dendritic regions, as well as the cell body. As expected from its role in synaptic vesicle release in motor neurons, we show that in Tm3, *cac* is required for normal light-evoked calcium responses in the axon terminals in layers Lo1 and Lo4. However, unexpectedly, the effects of Cac removal were substantially different across the other compartments of Tm3. In particular, Cac is not required in the axon terminals in layers M8 and M10. Thus, unlike in motor neurons where Cac is required at all synaptic boutons, in Tm3, some, but not all, output regions require this channel. Given that these arbors synapse onto distinct downstream partners, these molecular differences may be computationally significant. Additionally, we observed that loss of Cac reduces calcium responses in the cell body, an observation that also diverges from its role in motor neurons (*Worrell and Levine, 2008*). As the cell body neither makes nor receives synapses, Cac perhaps contributes to other processes such as calcium-mediated regulation of transcription or enzyme activity (*Catterall, 2011*). Finally, we were surprised to find that removal of Cac increased calcium responses to light specifically in the layer M5 arbor. The removal of a voltage-gated calcium channel from a compartment would not be expected to directly increase calcium signals. One explanation for this observed increase is that Cac was locally replaced with another voltage-gated calcium channel that perhaps has a lower activation threshold or slower inactivation, allowing more calcium to enter the layer M5 arbor upon stimulation. Indeed, wild-type calcium responses in the layer M5 arbor are much smaller than would be expected from the corresponding voltage signals (*Yang et al., 2016*). Perhaps Cac normally mediates this small response, but the channel that replaces it in *cac* mutant cells has different properties. Alternatively, loss of Cac throughout Tm3 may cause global changes in calcium channel regulation or expression and thus the phenotypes we observe may also be influenced by compensation. If this were the case, the compartment-specific effects of *cac* disruption on calcium signaling would reflect differential effects of compensation across subcellular compartments. Taken together, the effects of disrupting *cac* in Tm3 illustrate the complexity of neuronal calcium signaling and the diverse roles Cac plays in a central nervous system neuron.

## Outlook

FlpStop is a powerful tool for addressing questions of how specific genes establish the rich complement of neuronal responses that support computation. Importantly, because this approach manipulates the endogenous mechanisms that confer a neuron's response properties, it is fundamentally different from methods that activate or silence neurons through the expression of exogenous effectors. Thus, FlpStop will provide complementary and novel insights into nervous system function.

## Materials and methods

### Cloning of the FlpStop construct pFlpStop-attB-UAS-2.1-tdTom

First, the backbone of the construct containing the MHC splice acceptor, the stop codons, the SV40 terminator (*Stockinger et al., 2005*), and the series of F3 and wild-type FRT sites was synthesized by GenScript (New Jersey, USA) and delivered in the pUC57 vector flanked by EcoRI and HindIII sites. This backbone was cloned in between the two attB sites of the MiMIC cloning vector pBS-KS-attB1-2 (obtained from the Drosophila Genome Resource Center, Bloomington, IN) using EcoRI and HindIII.

A sequence containing a 5xUAS, hsp70 promotor, and TATA box were PCR amplified from a pUAST>stop>mCD8::GFP vector (obtained from Liqun Luo, Stanford University). During the amplification with primers

UAShsp70_5p_EcoRI: GCTTGAATTCCCTGCAGGTCGGAGTACT and

UAShs70_3p_XhoI: GTTAGAGCTCCCCAATTCCCTATTCAGAGTTCT, EcoRI and XhoI restriction sites were added to the DNA fragment. This fragment was then cloned into the growing FlpStop construct using the EcoRI and XhoI restriction sites.

Next, the Tub α 1 terminator and the SV40 terminator associated with the tdTomato element were added. First, these two transcriptional terminators flanked by NotI and FseI restriction sites on one side and an AgeI restriction site on the other were synthesized as a DNA fragment (gBlocks Gene Fragments, Integrated DNA Technologies). This fragment was cloned into the main vector between the NotI and AgeI sites. Finally, the tdTomato element was added (*Shaner et al., 2004*).

tdTomato flanked by NotI and FseI restriction sites was synthesized as a separate DNA fragment (gBlocks Gene Fragments, Integrated DNA Technologies). This fragment was cloned into the main vector between one of the FRT sites and tdTomato-associated SV40 terminator using NotI and FseI (*Figure 1—figure supplement 1a*).

## Cloning of the FlpStop construct pFlpStop-HDR-UAS-2.1-tdTom for CRISPR-HDR-based genome insertion

First, the backbone of the construct containing the 3XP3 promoter, the DsRed coding sequence, and the SV40 3' terminator flanked by loxP sites, ΦC31 attP sites, and homology arm multiple cloning sites (sequence identical to those in pHD-DsRed-attP from *Gratz et al., 2014*) was synthesized by GenScript (New Jersey, USA) and delivered in the pUC57-mini vector with unique MluI and KpnI sites separating the upstream attP and loxP sites. Next, the FlpStop cassette internal to and excluding the attB sites was PCR amplified from the construct pFlpStop-attB-UAS-2.1-tdTom with the following primers: 5'_Fwd_FlpStop_KpnI: GCGTAGTCGATCGCGGTACCGCAGGAATTCCCTGCAGG TC and 3'_Rev_FlpStop_KpnI:

TTATACGAAGTTATGGTACCGTCGACAAGCTTGGATCCG.

This fragment was cloned into the KpnI site of the pFlpStop-HDR-UAS-2.1-tdTom construct using Infusion cloning (Clontech Laboratories, Inc.). All inserts in the pUC57-mini vector were sequenced end to end (Sequetech DNA Sequencing Service) to ensure sequence identity (*Figure 1—figure supplement 2a*).

The full sequence of the pFlpStop-attB-UAS-2.1-tdTom and pFlpStop-HDR-UAS-2.1-tdTom plasmids are found in *Supplementary file 1* and *Supplementary file 2*, respectively, and the plasmids are available from Addgene (pFlpStop-attB-UAS-2.1-tdTom: https://www.addgene.org/88910/; pFlpStop-HDR-UAS-2.1-tdTom: https://www.addgene.org/89148/).

## Genotypes of flies

### *Figure 2* and *Figure 2—figure supplements 1* and *2*

Complementation tests

$ap^{UG035}$/CyO

$ap^{FlpStop\ ND}$/CyO

$ap^{FlpStop\ D}$/CyO

+/+

*ap* disruption and rescue time course

$y, w, hsFlp^{122}; ap^{UG035}/ap^{FlpStop\ ND}$

$y, w, hsFlp^{122}; ap^{UG035}/ap^{FlpStop\ D}$

Wing disc *ap* clones

+/+

$y, w, hsFlp^{122}/w$ or $Y; ap^{UG035}/ap^{FlpStop\ ND}; tubP\text{-}Gal4/ +$

### *Figure 3* and *Figure 3—figure supplement 1*

*ChAT, Gad1, Rdl,* and *VGlut* complementation tests

$y, w; ; ChAT^{FlpStop\ ND}$/TM3, Sb, Ser

$y, w; ; ChAT^{FlpStop\ D}$/TM3, Sb, Ser

$ChAT^{I9}$/TM3, Ser

Df(3R)ED2/TM3, Sb, Ser

$y, w; ; Gad1^{FlpStop\ ND}$/TM3, Sb, Ser

$y, w; ; Gad1^{FlpStop\ D}$/TM3, Sb, Ser

Df(3L)ED4341/TM3, Sb, Ser

$Gad1^{L352F}, e$ /TM3, Sb, Ser

$y, w; ; Rdl^{FlpStop\ ND}$/TM3, Sb, Ser

$y, w; ; Rdl^{FlpStop\ D}$/TM3, Sb, Ser

$Rdl^1$/TM3, Sb

Df(3L)Rdl-2/TM3, Sb

$y, w; VGlut^{FlpStop\ ND}$/CyO

$y, w; VGlut^{FlpStop\ D}$/CyO

*Df(2L)VGlut$^2$/CyO*

*para* and *cac* complementation tests

 *para$^{ts1}$/para$^{ts1}$*

 *y, w, para$^{FlpStop\ D}$/para$^{ts1}$*

 *y, w, para$^{FlpStop\ ND}$/para$^{ts1}$*

 *para$^{ts1}$/+*

 *cac$^{TS2}$/ cac$^{TS2}$*

 *y, w, cac$^{FlpStop\ D\ lock}$/ cac$^{TS2}$*

 *y, w, cac$^{FlpStop\ ND}$/ cac$^{TS2}$*

 *cac$^{TS2}$/+*

qRT-PCR

 *y, w, cac$^{FlpStop\ ND}$/FM7c; ; TM3, Sb/+*

 *y, w, cac$^{FlpStop\ D\ lock}$/FM7c; ; TM3, Sb/+*

 *y, w; ; Gad1$^{FlpStop\ ND}$ /TM3, Sb, Ser*

 *y, w; ; Gad1$^{FlpStop\ D}$/TM3, Sb, Ser*

 *y, w, para$^{FlpStop\ ND}$/FM7*

 *y, w, para$^{FlpStop\ D}$/FM7*

 *y, w; ; Rdl$^{FlpStop\ ND}$/TM3, Sb, Ser*

 *y, w; ; Rdl$^{FlpStop\ D}$/TM3, Sb, Ser*

 *y, w; ; Shal$^{FlpStop\ ND}$/Shal$^{FlpStop\ ND}$*

 *y, w; ; Shal$^{FlpStop\ D}$/Shal$^{FlpStop\ D}$*

## Figure 3—figure supplement 2

*y, w, cac$^{FlpStop\ ND}$/y, w, cac$^{FlpStop\ ND}$*

*w; ovo-Flp/CyO*

## Figure 4

*cac$^{FlpStop\ ND}$/+; UAS-Flp/UAS-GCaMP6f; Mi1$^{R19F01}$-Gal4/+*

*cac$^{FlpStop\ ND}$/w; UAS-Flp/UAS-GCaMP6f; Tm1$^{R74G01}$-Gal4/+*

*cac$^{FlpStop\ ND}$/+; UAS-Flp/UAS-GCaMP6f; T4$^{R54A03}$-Gal4/+*

## Figure 5 and Figure 5—figure supplement 1

*w/+; UAS-Flp, UAS-mCD8::GFP/ap$^{FlpStop\ ND}$; Mi1$^{R19F01}$-Gal4/+*

*y, w, cac$^{FlpStop\ ND}$/w; UAS-Flp, UAS-mCD8::GFP/+; Mi1$^{R19F01}$-Gal4/+*

*y, w/w; UAS-Flp, UAS-mCD8::GFP/+; Mi1$^{R19F01}$-Gal4/ChAT$^{FlpStop\ ND}$*

*y, w, para$^{FlpStop\ ND}$/w; UAS-Flp, UAS-mCD8::GFP/+; Mi1$^{R19F01}$-Gal4/+*

*y, w/w; UAS-Flp, UAS-mCD8::GFP/+; Mi1$^{R19F01}$-Gal4/Rdl$^{FlpStop\ ND}$*

*y, w/w; UAS-Flp, UAS-mCD8::GFP/+; Mi1$^{R19F01}$-Gal4/Shal$^{FlpStop\ ND}$*

*y, w/w; UAS-Flp, UAS-mCD8::GFP/+; Mi1$^{R19F01}$-Gal4/Gad1$^{FlpStop\ ND}$*

*y, w/w; UAS-Flp, UAS-mCD8::GFP/VGlut$^{FlpStop\ ND}$; Mi1$^{R19F01}$-Gal4/+*

*y, w/+; UAS-Flp/+; nSyb$^{R5710}$-Gal4/Gad1$^{FlpStop\ ND}$*

*y, w/+; UAS-Flp/VGlut$^{FlpStop\ ND}$; nSyb$^{R57C10}$-Gal4/+*

## Figure 6

Experimental: *cac$^{FlpStop\ ND}$/Y; UAS-GCaMP6f/UAS-Flp; Tm3$^{R13E12}$-Gal4/tubP-Gal80$^{ts}$*

No-Flp control: *cac$^{FlpStop\ ND}$/Y; UAS-GCaMP6f/+; Tm3$^{R13E12}$-Gal4/tubP-Gal80$^{ts}$*

## Figure 7

Experimental: *cac$^{FlpStop\ ND}$/Y; UAS-GCaMP6f/UAS-Flp; Tm3$^{R13E12}$-Gal4/tubP-Gal80$^{ts}$*

No-Flp control: *cac$^{FlpStop\ ND}$/Y; UAS-GCaMP6f/ +; Tm3$^{R13E12}$-Gal4/tubP-Gal80$^{ts}$*

Heterozygous control: *cac$^{FlpStop\ ND}$/w; UAS-GCaMP6f/UAS-Flp; Tm3$^{R13E12}$-Gal4/tubP-Gal80$^{ts}$*

 Note that in these experiments the no-Flp control should neither disrupt *cac* nor express tdTomato but otherwise contains all other transgenes, thereby controlling for their presence. The heterozygous control should invert *cac$^{FlpStop\ ND}$* and express tdTomato; however, as it carries one wild-type *cac* allele, *cac* function in Tm3 should be preserved. This controls for any potential effects of

Flp and tdTomato expression on Tm3. These calcium imaging experiments were performed in flies bearing a wild-type copy of *white*. The *cac^FlpStop ND* chromosome was obtained by recombining off the *yellow* (*y*) and *white* (*y*) mutations that were present in the original MiMIC stock (*Venken et al., 2011a*).

Note that the FlpStop allele was placed in trans to another allele, as is standard practice to avoid the effects of background recessive mutations on a given chromosome. This also circumvented any potential interactions between the FRT sites in flies homozygous for the FlpStop allele.

## Complete stock list

Initial MiMIC lines (*Venken et al., 2011a*; *Nagarkar-Jaiswal et al., 2015a*):

 *y, w; ap^MI01996/CyO*
 *y, w, cac^MI02836*
 *y, w; ; Gad1^MI09277/ TM3, Sb*
 *y, w, para^MI08578*
 *y, w; ;Rdl^MI02620/ TM3, Sb*
 *y, w; ; ChAT^MI04508/ TM3, Sb*
 *y, w; Shal^MI00446*
 *y, w; VGlut^MI04979/ CyO*

Apterous disruption and rescue time course:

 *y, w, hsFlp^122* (*Golic and Lindquist, 1989*)
 *ap^UG035/CyO* (*Cohen et al., 1992*)
 *+; ap^FlpStop ND/ CyO* (this study)
 *y, w; ap^FlpStop D/ CyO* (this study)

Apterous wing disc immunolabeling:

 *S/CyO, Kr-GFP* (obtained from T Mosca)
 *ap^FlpStop ND/CyO, Kr-GFP* (this study)
 *ap^FlpStop D/CyO, Kr-GFP* (this study)
 *y, w, hsFlp^122* (*Golic and Lindquist, 1989*)
 *tubP-Gal4/TM6B, Tb* (*Lee and Luo, 1999*)

Complementation tests:

 *ap^UG035/CyO* (*Cohen et al., 1992*)
 *cac^TS2/cac^TS2* (obtained from R. Ordway) (*Kawasaki et al., 2000*)
 *ChAT^I9/TM3, Ser* (obtained from BDSC)
 *Df(3R)ED2/TM3, Sb, Ser* (obtained from BDSC)
 *Df(3L)ED4341/TM3, Sb, Ser* (obtained from BDSC)
 *Gad1^L352F, e/TM3, Sb, Ser* (obtained from BDSC)
 *para^ts1/para^ts1* (obtained from B. Ganetzky) (*Suzuki et al., 1971*)
 *Rdl^1/TM3, Sb, Ser* (obtained from BDSC) (*Ffrench-Constant et al., 1991*)
 *Df(3L)Rdl-2/TM3, Sb, Ser* (obtained from BDSC) (*Ffrench-Constant et al., 1991*)
 *Df(2L)VGlut^2/CyO* (obtained from A DeAntonio) (*Daniels et al., 2006*)
 *y, w; ap^FlpStop ND/ CyO* (this study)
 *y, w; ap^FlpStop D/ CyO* (this study)
 *y, w; ; ChAT^FlpStop ND/TM3, Sb, Ser* (this study)
 *y, w; ; ChAT^FlpStop D/TM3, Sb, Ser* (this study)
 *y, w; VGlut^FlpStop ND/SM6a* (this study)
 *y, w; VGlut^FlpStop D/SM6a* (this study)
 *y, w, cac^FlpStop ND/FM7c* (this study)
 *y, w, cac^FlpStop D lock/FM7c* (this study)
 *y, w; ; Gad1^FlpStop ND/TM3, Sb, Ser* (this study)
 *y, w; ; Gad1^FlpStop D/TM3, Sb, Ser* (this study)
 *y, w, para^FlpStop ND/FM7* (this study)
 *y, w, para^FlpStop D/FM7* (this study)
 *y, w; ; Rdl^FlpStop ND/TM3, Sb, Ser* (this study)
 *y, w; ; Rdl^FlpStop D/TM3, Sb, Ser* (this study)

qRT-PCR:

 *y, w, cac^FlpStop ND/FM7c; ; TM3, Sb/+* (this study)

y, w, cac[FlpStop D lock]/FM7c; ; TM3, Sb/+ (this study)
y, w; ; Gad1[FlpStop ND]/TM3, Sb, Ser (this study)
y, w; ; Gad1[FlpStop D]/TM3, Sb, Ser (this study)
y, w, para[FlpStop ND]/FM7 (this study)
y, w, para[FlpStop D]/FM7 (this study)
y, w; ; Rdl[FlpStop ND]/TM3, Sb, Ser (this study)
y, w; ; Rdl[FlpStop D]/TM3, Sb, Ser (this study)
y, w; ; Shal[FlpStop ND]/Shal[FlpStop ND] (this study)
y, w; ; Shal[FlpStop D]/Shal[FlpStop D] (this study)

Germline inversion of cac[FlpStop ND]:
w; ovo-Flp/CyO (obtained from BDSC)

Fluorescence reporter testing:
Mi1[R19F01]-Gal4 (obtained from BDSC) (*Jenett et al., 2012*)
Tm1[R74G01]-Gal4 (obtained from BDSC) (*Jenett et al., 2012*)
T4[R54A03]-Gal4 (obtained from BDSC) (*Jenett et al., 2012*)
nSyb[R57C10]-Gal4 (obtained from BDSC) (*Jenett et al., 2012*)
UAS-Flp/CyO (obtained from BDSC)
UAS-mCD8::GFP/CyO (obtained from L Luo) (*Lee and Luo, 1999*)
cac[FlpStop ND] (this study)
UAS-GCaMP6f/CyO (obtained from BDSC) (*Chen et al., 2013*)
+; ap[FlpStop ND]/ CyO (this study)
y, w, cac[FlpStop ND]/FM7c (this study)
y, w; ; ChAT[FlpStop ND]/TM3, Sb, Ser (this study)
y, w, para[FlpStop ND]/FM7 (this study)
y, w; ; Rdl[FlpStop ND]/TM3, Sb, Ser (this study)
y, w; ; Shal[FlpStop ND]/ TM3, Sb, Ser (this study)
y, w; ; Gad1[FlpStop ND]/TM3, Sb, Ser (this study)
y, w; VGlut[FlpStop ND]/SM6a (this study)

Tm3 *cac* FlpStop:
Tm3[R13E12-]Gal4 (obtained from BDSC) (*Jenett et al., 2012*)
tubP-Gal80[ts] (obtained from BDSC) (*McGuire et al., 2003*)
UAS-Flp/CyO (obtained from BDSC)
UAS-GCaMP6f/CyO (obtained from BDSC) (*Chen et al., 2013*)
cac[FlpStop ND] (this study)

## Fly husbandry

Unless otherwise stated, all flies were raised on standard molasses food at 25°C on a 12/12 hr light-dark cycle. Parental crosses were flipped into new vials approximately every two days. For Tm3 *cac* FlpStop experiments, flies were raised at 18°C until they were wandering third instar larvae. They were then raised at 29°C. As our Tm3 driver is expressed in other cells during larval stages (*Jenett et al., 2012*), removal of *cacophony* within the pattern defined by this driver line was developmentally lethal. To circumvent this lethality, we incorporated the a *tub-Gal80[ts]* transgene (ubiquitously expressed temperature-sensitive repressor of Gal4 (*McGuire et al., 2003*)) and used the temperature shifts stated above to restrict Gal4 activity to late pupal and adult stages.

## Creation of FlpStop transgenic flies

The FlpStop attB donor plasmid, pFlpStop-attB-UAS-2.1-tdTom, was injected into flies containing the MiMIC insertion and a ΦC31 integrase-expressing transgene. Flies were screened for insertion of the FlpStop cassette by the loss of *yellow* rescue from the MiMIC construct. The orientation of the cassette was then confirmed by PCR using one primer within the Minos backbone of the MiMIC construct

Orientation-MiL-F: GCGTAAGCTACCTTAATCTCAAGAAGAG (*Venken et al., 2011a*)
and two other primers with homology to the two ends of the FlpStop cassette:
FRTspacer_5p_rev: AAATGGTGCAAAGAGAAGTTCC
FRTspacer_3p_for: ACAATCCAGCTACCATTCTGC

BestGene Inc. (Chino, CA) performed the embryo injections for, established, and PCR-verified the majority of the FlpStop stocks.

## Germline inversion of the *cac*^*FlpStop* allele

For *cac*, only the non-disrupting allele was isolated from the first round of injections. To enable assessment of the mutagenicity of *cac*^*FlpStop*, an allele in the disrupting orientation was made by expressing Flp recombinase in the germline using *ovo-Flp* (*Figure 3—figure supplement 2*). Cassette inversion was then confirmed using four different PCR reactions that together could distinguish the full cassette inversion from a partial inversion (only one FRT reaction) or the original, uninverted cassette. The distinguishing PCR primer pairs were: Orientation-MiL-R and tdTomato_5p_Rev, Orientation-MiL-R and FRTspacer_5p_rev, Orientation-MiL-F and SV40_for, and Orientation-MiL-F and FRTspacer_3p_for.

### Primer sequences:
Orientation-MiL-F: GCGTAAGCTACCTTAATCTCAAGAAGAG (*Venken et al., 2011a*)
Orientation-MiL-R: CGCGGCGTAATGTGATTTACTATCATAC (*Venken et al., 2011a*)
FRTspacer_5p_rev: AAATGGTGCAAAGAGAAGTTCC
FRTspacer_3p_for: ACAATCCAGCTACCATTCTGC
tdTomato_5p_rev: CCCTTGGTCACTTTCAGCTT
SV40_for: GAAGACCCCAAGGACTTTCC

## Complementation tests

### Lethality

Null mutant stocks or FlpStop alleles were established using compatible balancer chromosomes: *FM7* (X); *SM6a, Cy* or *CyO* (second); or *TM3, Ser, Sb* or *TM3, Sb* or *TM3, Ser* (third). Crosses were always made with heterozygous animals carrying the same balancer. Upon eclosion, an adult male and female progeny were counted and scored for the presence or absence of the balancer chromosome. Significance was assessed using a one-proportion z-test against the null hypothesis that the survival of *FlpStop/Null* flies will be 33% based on the predicted inheritance for all of the crosses performed.

### Paralysis

Groups of 5 flies were placed into empty plastic vials sealed by a cotton plug. Vials were then submerged in a heated water bath for 2 min. The vials were then removed from the water bath, and the number of flies paralyzed at the bottom of the vial was quickly counted. Paralysis testing for *cac* was performed at 37°C and 40°C. Paralysis testing for *para* was performed at 33°C or 35°C. For each assay, each group of flies was always tested at both temperatures. They were allowed to recover on standard molasses food for at least 20 min between tests. Significance was assessed using a two-tailed Fisher's exact test to compare each genotype to the *TS/+* control.

### Wing phenotypes

Each wing was scored independently for the severity of the wing phenotype on an established scale from 1 to 5 (*Gohl et al., 2008*). For *Figure 2b*, female flies lightly anesthetized with $CO_2$ were photographed using a CCD camera mounted on a dissection microscope.

## *apterous* disruption and rescue experiments

*y, w, hsFlp*^*122*/+ or Y; *ap*^*UG035*/*ap*^*FlpStop ND* and *y, w, hsFlp*^*122*/w or Y; *ap*^*UG035*/*ap*^*FlpStop D* flies were heat-shocked once at 37°C for 1 hr at the indicated age (1 to 7 days where 1 day is 24 hr after the parents were first placed into a new vial to begin laying eggs). As a control, flies of the same genotypes were not heat-shocked. Flies that eclosed within 24 hr after the first fly eclosed from a given vial were scored for their wing phenotypes. Both male and female flies were scored. Each wing was scored independently for the severity of the wing phenotype on an established scale from 1 to 5 (*Gohl et al., 2008*). For *Figure 2d*, female flies lightly anesthetized with $CO_2$ were photographed using a CCD camera mounted on a dissection microscope.

## Wing disc *ap* clones

*y, w, hsFlp*$^{122}$*/w* or *Y; ap*$^{UG035}$*/ap*$^{FlpStop\ ND}$*; tubP-Gal4/ +* flies were heat-shocked at 37°C for 15 to 20 min during early development (first or second instar). As a control, flies of the same genotype were not heat-shocked. Wing discs from wandering third instar larvae were examined (see Immunolabeling).

## qRT-PCR

For each biological replicate, RNA was extracted from the brains of approximately 10 adult flies. Brains were microdissected out of the flies' heads in PBS. Brains were then homogenized in a solution containing beta-mercaptoethanol, and RNA was extracting using an RNeasy Mini Kit (Qiagen). All of the RNA samples were checked for quality by bioanalysis to confirm the presence of both 18S and 28S ribosomal peaks and the lack of evidence of RNA degradation (RNA integrity number > 5). QC Bioanalysis was performed by the Stanford PAM facility using an Agilent 2100 Bioanalyzer. cDNA was made from the RNA samples using the SuperScript VILO Master Mix (Thermo Fisher Scientific), and this cDNA solution was used directly in subsequent qRT-PCR reactions. Four technical replicates were run for each of the biological replicates of each genotype, and the mean of those four readings was used to estimate transcript levels for each individual biological replicate. For all of the qRT-PCR experiments, *Gapdh2* was used as a reference for normalization. Cycle threshold (Ct) values were obtained from the Eppendorf software. The Delta-Delta Ct method was used to calculate relative transcript levels: % Transcript difference = $2^{\wedge}((Ct\ ND\ target\ gene - Ct\ ND\ reference\ gene) - (Ct\ D\ target\ gene - Ct\ D\ reference\ gene))$. An unpaired, two-tailed, two-sample Student's t-test was applied to the delta Ct values obtained for the biological replicates to assess significance. For presentation purposes, all observations were normalized to the mean transcript level of the non-disrupting genotype.

All primers were tested for primer efficiency using serial dilution of control Canton-S cDNA. Primers were designed to amplify a product that spanned the two exons flanking the intron containing the FlpStop cassette. For each locus, three primer pairs were initially designed and tested with a 5-fold serial dilution of Canton-S cDNA. The primer pair with the efficiency closer to 2 (idealized PCR doubling efficiency) was used for the quantification experiments on the FlpStop genotypes. Measured efficiency values are noted next to each primer pair below.

## List of qRT-PCR primers

*Gapdh2* reference primers (efficiency: 1.94)
GAPDH2-qPCR-F: CGTTCATGCCACCACCGCTA
GAPDH2-qPCR-R: CCACGTCCA TCACGCCACAA

*cac* (efficiency: 2.48)
Cac_Exons2021_for1: ACTTTGGATGGAATTCGGGTC
Cac_Exons2021_rev1: TTCGAATCGTGTATCCTTGACG

*Gad1* (efficiency: 2:16)
Gad1_Exon45_for3: AAGCATCGTCATCCCAGATTT
Gad1_Exon45_rev3: TCCTTGAAATGGATCGTGGAG

*para* (efficiency: 2.10)
para_Exon345_for1: GAGGTGCCGCAATATGGTC
para_Exon345_rev1: CACACCCTGTTCAAGTGTAGG

*Rdl* (efficiency: 1.96)
Rdl_Exon34Junc_for3: TATCGATCAGCTCGCTCTCA
Rdl_Exon34Junc_rev3: CTATACGCTAAACGAGGATCGG

*Shal* (efficiency: 2.02)
Shal_Exon23_for2: AGGGAGTTCTTCTACGACGAG
Shal_Exon23_rev2: CCGTCCGGTAGTAGTTCAGT

## Immunolabeling

To image the adult brain, flies approximately 1 day ($T4^{R54A03}$-Gal4), 5–7 days ($Mi1^{R19F01}$-Gal4, $Tm1^{R74G01}$-Gal4, $nSyb^{R57C10}$-Gal4) (*Figures 4* and *5*), or 10–12 days ($Tm3^{R13E12-}$Gal4) (*Figure 6*) post-eclosion were dissected, fixed, and stained using standard methods. To observe wing discs, wandering third instar larvae were dissected, fixed, and stained using standard methods (*Morante and Desplan, 2011*). The primary antibodies used in this study were anti-GFP (chicken, Abcam, 1:2000, RRID:AB_300798), anti-Bruchpilot (nc82, mouse, Developmental Studies Hybridoma Bank, 1:30, RRID:AB_2314867), anti-DsRed (rabbit, Clontech, 1:500, RRID:AB_10013483), anti-Apter-ous (rabbit, gift from D Bieli, University of Basal, Switzerland, 1:1000), and anti-RFP (chicken, 1:1000 Millipore, RRID:AB_91496). The anti-GFP antibody recognized both GFP and GCaMP6f. The anti-DsRed and anti-RFP antibodies both recognized tdTomato. Adult brains and larval wing discs were imaged using a Leica TCS SP8 confocal microscope (Bensheim, Germany) using either a Leica HC PL APO 20x/0.70-NA or a Leica HCX PL APO 40x/1.25–0.75-NA oil immersion objective. Confocal images were rendered in three dimensions using Imaris (Bitplane), adjusted using cropping and thresholding tools in Photoshop (Adobe), and assembled into figures using Illustrator (Adobe).

## Quantification of FlpStop cassette inversion efficiency

To quantify the FlpStop cassette inversion efficiency, we performed cell body counting using Imaris (Bitplane) to count the number of GFP-positive neurons that were also tdTomato-positive. In the 3D 'Surpass' visualization mode, we manually placed 'spots' on the middle of each GFP-positive cell body (GCaMP6f in *Figure 4 and 6*, mCD8::GFP in *Figure 5*). Spots labeling GFP-positive cells were placed while blinded to the tdTomato channel. Next, 'spots' marking the center of tdTomato posi-tive cell bodies were placed while blinded to the GFP channel. Colocalization between GFP and tdTomato spots was then obtained using the 'Colocalize Spots' function. For experiments labeling Mi1, Tm1, and Tm3, a threshold distance of less than 3.5 µm between the two spots was used as the cutoff for colocalization. For the T4 analysis this threshold was lowered to 2.5 µm since the T4 cell bodies were more closely clustered together. After running the colocalization function, the analysis was checked by eye and any obvious false positive or false negative errors were corrected by chang-ing either the location of a spot, removing a spot, or adding a spot. The main analysis work flow fol-lowed the instructions laid out in this guide: Imaris Guide, Cell Counting, Queensland Brain Institute's Advance Microimaging and Analysis Facility: http://web.qbi.uq.edu.au/microscopy/imaris-counting-cells-on-slide-scanner-images/

## In vivo imaging

Flies were cold anaesthetized, positioned in a fly-shaped hole cut in steel foil such that their heads were tilted forward approximately 90° to expose the back of the head capsule above the foil while leaving most of the retina below the foil, and then affixed in place with UV-cured glue (NOA 68T from Norland Products Inc.). The brain was exposed by removing the overlying cuticle and fat bodies with fine forceps, and an oxygenated saline-sugar solution (*Wilson et al., 2004*) was perfused over the fly. The saline composition was as follows: 103 mM NaCl, 3 mM KCl, 5 mM TES, 1 mM $NaH_2PO_4$, 4 mM $MgCl_2$, 1.5 mM $CaCl_2$, 10 mM trehalose, 10 mM glucose, 7 mM sucrose, and 26 mM $NaHCO_3$. The pH of the saline equilibrated near 7.3 when bubbled with 95% $O_2$/5% $CO_2$. Neurons were imaged with a Leica TCS SP5 II microscope with a Leica HCX APO 20X/1.0-NA water immersion objective (Leica). For two-photon imaging, a pre-compensated Chameleon Vision II femtosecond laser (Coherent, Inc.) tuned to a wavelength of 920 nm was used to excite the sample (5–15 mW of power at the stage). Emitted photons from GCaMP6f were collected with a 525/50 nm filter. Visually evoked calcium responses were acquired at a constant frame rate of 38.9 Hz using a frame size of 200 × 20 pixels, 15X digital zoom, a line scan rate of 1400 Hz, and bidirectional scanning. Imaging time per fly never exceeded 1.5 hr. Tm3 calcium signals were imaged in adult flies 10 to 12 days after eclosion. For one-photon imaging, a 488 nm diode laser was used to excite GCaMP6f, and emitted photons with wavelengths between 496 nm and 537 nm were collected. A 543 nm diode laser was used to excite tdTomato, and emitted photons with wavelengths between 557 nm and 661 nm were collected.

## Visual stimulation

Visual stimuli were generated with custom-written software using C++ and OpenGL (available on GitHub at https://github.com/ClandininLab/2pstim) and presented using a digital light projector as described previously (*Clark et al., 2011*). The visual stimulus was projected onto a coherent fiber optic bundle that then re-projected onto a rear-projection screen positioned approximately 4 cm anterior to the fly that spanned 80° of the fly's visual field horizontally and 50° vertically. Immediately prior to being projected onto the screen, the stimulus was filtered with a 482/18 nm bandpass filter as well as an ND 0.5 neutral density filter so that it could not be detected by the microscope PMTs. The stimulus was refreshed at 240 Hz and had a radiance of approximately 30 mW·sr$^{-1}$·m$^{-2}$. The imaging and the visual stimulus presentation were synchronized as described in *Freifeld et al. (2013)*. Following this procedure, the time of stimulus onset relative to the start of imaging varied within one stimulus frame (8.33 ms). To compensate for this, the average delay was measured (6.25 ms), and all imaging data was shifted in time by this delay.

The visual stimuli used were:

*2 s flash search stimulus:* alternating full contrast light and dark flashes, each 2 s in duration, were presented within the central region of the screen while the remainder of the screen was kept dark. The distance between the edges of the flashing stimulus and the edges of the screen spanned approximately 15° of the fly's visual field. This stimulus was presented for 2400 imaging frames (62 s) per field of view.

*25 ms light and dark flashes:* 25 ms light and dark flashes, with 1500 ms of gray between the flashes, were presented over the entire screen. The light and dark flashes alternated with each presentation. The Michelson contrast of the flashes relative to the gray was 0.5. This stimulus was presented for 6600 imaging frames (170 s) per field of view. Tm3's initial calcium response to dark flashes was minimal (*Yang et al., 2016*) and was not altered by Cac removal (data not shown). Therefore, only responses to light flashes were further analyzed.

Both stimuli were presented to all cells.

## In vivo imaging data analysis

Raw images in each time series were aligned in x and y coordinates in ImageJ (NIH) using a macro based on the plugin Turboreg (http://bigwww.epfl.ch/thevenaz/turboreg/) and then further processed in MATLAB (Mathworks, MA). Regions of interest (ROIs) around individual arbors or cell bodies were manually selected in the time series-averaged image. For each imaging frame within the time series, intensity values for the pixels within each ROI were averaged and the mean background value (the average intensity in a region of the image without cells) was subtracted. To correct for bleaching, the time series for each ROI was fit with the sum of two exponentials, and in the calculation of $\Delta F/F = (F(t) - F_0)/F_0$, the fitted value at each time t was used as $F_0$. For the 2 s flash search stimulus, all imaging frames were used to compute the fit, thereby placing $\Delta F/F = 0$ at the mean response after correction for bleaching. For the 25 ms light and dark flashes off of gray stimulus, only imaging frames that fell in the last 25% of the gray period were used to fit the bleaching curve; this places $\Delta F/F = 0$ at the mean baseline the cell returns to after responding to the flash instead of at the mean of the entire trace. The stimulus-locked average was computed for each ROI by reassigning the timing of each imaging frame to be relative to the stimulus transitions (dark to light or light to dark for the 2 s flash search stimulus, gray to light or gray to dark for the flashes off of gray) and then computing a simple moving average with a 25 ms averaging window and a shift of 8.33 ms. This effectively resampled our data from 38.9 Hz to 120 Hz with 3-point boxcar smoothing but did not otherwise distort the fluorescence signal. As the screen on which the stimulus was presented did not span the fly's entire visual field, only a subset of imaged ROIs actually observed the stimulus. Both the 2 s flash search stimulus and the 25 ms light and dark flashes stimulus were presented to each ROI. Responding ROIs were identified as those responding to the 2 s flash search stimulus. All of the responses also were examined manually. All traces are presented as the mean ±1 SEM across all of the responding ROIs of the moving average response to all trials of each ROI.

Peak $\Delta F/F$ for each ROI was computed as follows:

For the responses to the 25 ms light and dark flashes off of gray stimulus, the peak response to the light flash (peak $\Delta F/F$) was the $\Delta F/F$ value farthest from zero in the expected direction of the initial response (calcium increase). Each ROI's peak $\Delta F/F$ was then normalized to the mean peak $\Delta F/F$

of the no-Flp control and presented as mean ±1 SEM. One-way ANOVAs comparing the experimental and two control genotypes were performed. Post-hoc Student's two-sample t-tests between the experimental and each of the two control genotypes were performed and Bonferroni-corrected for multiple comparisons.

## Quantification of the percentage of neuronal genes with a MiMIC site in a coding intron

A list of *D. melanogaster* neuronal genes was obtained by using FlyBase's QueryBuilder tool to search for genes expressed in the nervous system. The DataClass field was set to 'Expression Patterns', the Body Part/Tissue field was set to 'nervous system', and the species filter was set to 'Dmel'. This yielded ~2000 neuronal genes. Based on this list of neuronal genes and the list of MiMIC insertions (http://flypush.imgen.bcm.tmc.edu/pscreen/files/MI-list-2016-06-30.xlsx), we identified the subset of neuronal genes that had at least one MIMIC site in a coding intron.

## Acknowledgements

We thank T Mosca and members of the Clandinin Laboratory for helpful discussion and comments on this manuscript. We thank D Bieli (University of Basel), M Müller (University of Basel), L Luo (Stanford University), T Mosca (Stanford University), A DeAntonio (Washington University in St. Louis), B Ganetzky (University of Wisconsin), R Ordway (Pennsylvania State University), the Drosophila Genome Resource Center (DGRC), and the Bloomington Drosophila Stock Center (BDSC) for reagents. We thank R Volkan for an unpublished qRT-PCR primer sequence; M Bennett, C Bennett, T Li, M Fu, and S Liddelow (Stanford University) for advice about qRT-PCR experiments; and B Barres (Stanford University) for use of a qRT-PCR machine. We thank the Harvard Image and Data Analysis Core (IDAC) for access to Imaris software for analysis of immunostaining. YEF was supported by a National Science Foundation Fellowship. HHY was supported by a Stanford Graduate Fellowship and a Stanford Interdisciplinary Graduate Fellowship. DMG was supported by a Ruth L Kirschstein NRSA Postdoctoral Fellowship (F32 EY020040) from the National Eye Institute. This work was funded by R01 EY022638 and U01 MH109119 (to TRC).

## Additional information

### Funding

| Funder | Grant reference number | Author |
| --- | --- | --- |
| National Eye Institute | R01 EY022638 | Thomas R Clandinin |
| National Institute of Mental Health | U01 MH109119 | Thomas R Clandinin |
| National Science Foundation | | Yvette E Fisher |
| Stanford University School of Medicine | | Helen H Yang |
| National Eye Institute | F32 EY020040 | Daryl M Gohl |

The funders had no role in study design, data collection and interpretation, or the decision to submit the work for publication.

### Author contributions

YEF, designed the study, developed and created the FlpStop transgenic stocks, performed the apterous developmental time course, performed analysis of mosaic clones, performed genetic complementation testing and qRT-PCR experiments, performed confocal imaging of immunostained adult fly brains, analyzed data, and wrote the manuscript; HHY, performed the apterous developmental time course, performed analysis of mosaic clones, assisted with genetic complementation testing and qRT-PCR experiments, performed confocal imaging of immunostained adult fly brains, performed two-photon imaging of Tm3 neurons, analyzed data, and wrote the manuscript; JI-B, created the CRISPR-HDR-compatible FlpStop plasmid; MX, performed two-photon imaging of Tm3 neurons, analyzed data, and assisted with writing of the manuscript; DMG, designed the study and

helped to develop the FlpStop construct; TRC, designed the study, performed analysis of mosaic clones, performed confocal imaging of immunostained adult fly brains, and wrote the manuscript

**Author ORCIDs**

Helen H Yang, http://orcid.org/0000-0001-5140-9664

Thomas R Clandinin, http://orcid.org/0000-0001-6277-6849

---

## Additional files

**Supplementary files**

• Supplementary file 1. GenBank file of the full sequence of pFlpStop-attB-UAS-2.1-tdTom.

• Supplementary file 2. GenBank file of the full sequence of pFlpStop-HDR-UAS-2.1-tdTom.

---

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
