## [Decision Letter]

Thank you for submitting your article "FlpStop: a tool for conditional gene control in *Drosophila*" for consideration by *eLife*. Your article has been favorably evaluated by K VijayRaghavan as the Senior Editor and three reviewers, one of whom is a member of our Board of Reviewing Editors. The reviewers have opted to remain anonymous.

The reviewers have discussed the reviews with one another and the Reviewing Editor has drafted this decision to help you prepare a revised submission.

Summary:

This manuscript by Fisher et al. presents a new method ("FLP-Stop") that uses FLP-mediated recombination to accomplish post-mitotic disruption of targeted genes in cell types of interest in *Drosophila*. The authors generate a FRT-stop-FRT cassette with a red reporter that integrates into existing MiMIC sites in the genome to allow FLP-mediated gene disruption (or rescue) and labeling of mutant cells. The manuscript assesses the effectiveness of this approach and performs a number of critical validation experiments.

The manuscript is well-written and generally easy to read. The method is ingenious, and has the potential to be a very useful technique for scientists working in *Drosophila* – particularly in the neurosciences. One area in which the manuscript could be improved is by providing a more systematic assessment of the efficiency of Flp-mediated conversion and discussion of FlpStop's disadvantages as well as its advantages.

Essential revisions:

1) Why do the three of the seven genes targeted not generate null alleles? This is an important point for expanded discussion and hints at limitations of the approach. Is there anything in the insert location in *ChAt* that may suggest why disruption did not work? Was the insert incapable of flipping (testable by expression of tdTomato) or simply skipped? A better understanding of why the approach did not work in some cases may allow one to produce better reagents for new genes of interest.

2) Quantitative assessment of the efficiency of Flp for the examples of FlpStop usage discussed in the manuscript would be useful and this data should be included either in the text or in a table. The "flipping" frequency across loci is not well documented and seems variable (Figure 5). The authors report ~80% of red-green cells using *cac^FLPstop^* and M1-Gal4, is this number a function of the locus or of the GAL4/FLP combination? What is the frequency for other loci using the same Gal4/FLP combo and for the same locus using different Gal4s? (There is a hint of this answer in the legend of Figure 5—figure supplement 1; this should be addressed head on and quantified). Moreover, Figure 5 panel B seems to show quite a good number of cells that are green but not red and possibly a number of cells that are red but not green (e.g. in the calyx). How is this second population explained? This figure will benefit from higher resolution images and better quantification.

3) The advantages and disadvantages of FlpStop should be thoroughly discussed. With regard to the utility of the FlpStop toolkit to a potential end-user: a clear constraint is its dependence on the MiMIC insertion lines. The authors assert that MiMIC insertions "allow access to the coding introns of approximately 46% of neuronal genes" (subsection “FlpStop enables a diversity of applications”, first paragraph) and that "FlpStop alleles can be created that target any desired genomic locus using CRISPR/Cas9." The first statement seems likely to be a serious overestimate (using the authors methodology, I come up with MIMIC insertions into 23% of "neuronal genes"). While the second assertion is true, it would be more compelling if the manuscript presented and validated a FlpStop construct that was CRISPR/Cas ready if such a reagent has been made. The need for a null allele for all non-X-linked genes should also be addressed. The general possibility of using the FlpStopD insertion itself is not discussed, but has the obvious disadvantage of unveiling second site mutations on the same chromosome. What other general strategies (e.g. deficiencies) exist, and how useful might they be?

4) Subsection “FlpStop can produce conditional null alleles”. This section demonstrates at length that the FlpStop cassettes can do what MiMIC cassettes have already been shown to do – namely disrupt gene expression. While it is necessary to show that FlpStop works, this information would be of greater interest if it compared the efficacy of gene disruption using FlpStop with that of a MiMIC insertion in the disrupting orientation. One design difference between the MiMIC and FlpStop constructs is that the latter has transcriptional terminators, and it would be interesting to know if they are actually useful (i.e. do FlpStopDs reduce transcript levels more than the original MiMICs, when the latter are in a disrupting orientation?)

---

## [Author Response]

*Essential revisions:*

*1) Why do the three of the seven genes targeted not generate null alleles? This is an important point for expanded discussion and hints at limitations of the approach. Is there anything in the insert location in ChAt that may suggest why disruption did not work? Was the insert incapable of flipping (testable by expression of tdTomato) or simply skipped? A better understanding of why the approach did not work in some cases may allow one to produce better reagents for new genes of interest.*

We agree that this is an important topic. Therefore, we have expanded on this in the Results and Discussion sections and have added a supplemental figure to address this issue more directly.

All of the alleles are capable of efficient flipping, as revealed by tdTomato expression. This tdTomato inversion data is now included in Figure 5 and Figure 5—figure supplement 1, even for the alleles that were not nulls when in the disrupting orientation.

Of the eight targeted genes, the disrupting alleles of five were null, two were hypomorphs, and one did not disrupt gene function. We have some ideas about why these three alleles did not completely disrupt gene function:

*VGlut: VGlut^FlpStop D^* (the disrupting allele) was a hypomorph, as complementation testing revealed reduced viability but not the full lethality expected for a null allele. The *VGlut* FlpStop insertion lies between exons 3 and 4, and consistent with our observation, a previously characterized deletion of exons 1-3 of *VGlut* creates a viable allele that only partially reduces *VGlut* function (Daniels et al. 2006). Thus, in this case, we feel that the FlpStop cassette worked as it should have, but the location of insertion precluded the generation of a null allele. This data is now included in Figure 3—figure supplement 2 and discussed in both the Results and Discussion sections (subsections “FlpStop can produce conditional null alleles”, first paragraph and “The current FlpStop tool kit”, first paragraph).

*Shal*: Null alleles of *Shal* are homozygous viable, so qRT-PCR was used to assess gene disruption. This data revealed that the disrupting allele produced a 50% reduction in *Shal* mRNA but did not eliminate it. We speculate that the SA-STOP may be less efficient at blocking transcription and splicing when it occurs very early in the coding region of a gene of interest; however, we do not know with certainty why this allele was hypomorphic. This qRT-PCR data has been added to the paper (Figure 3—figure supplement 2) and is discussed in the Results and Discussion sections (subsections “FlpStop can produce conditional null alleles”, third paragraph and “The current FlpStop tool kit”, first paragraph).

*ChAT*: The *ChAT* FlpStop allele was the only one that failed to show evidence of gene disruption. Complementation testing revealed that *ChAT^FlpStop D^* was completely viable. *ChAT* is imbedded in a complex genetic locus where the coding region of *VAChT* is actually nested within the first intron of *ChAT*, and it has been proposed that both transcripts arise from a common transcript by differential RNA processing (Kitamoto et al., 1998). We speculate that complex, locus-specific regulation may make this gene it insensitive to the mutagenic signals imbedded within the FlpStop cassette. The complementation testing data has been included in Figure 3—figure supplement 2, and *ChAT* is further discussed in the Results and Discussion sections (subsections “FlpStop can produce conditional null alleles”, first paragraph and “The current FlpStop tool kit”, first paragraph).

*2) Quantitative assessment of the efficiency of Flp for the examples of FlpStop usage discussed in the manuscript would be useful and this data should be included either in the text or in a table. The "flipping" frequency across loci is not well documented and seems variable (Figure 5). The authors report ~80% of red-green cells using cac^FLPstop^ and M1-Gal4, is this number a function of the locus or of the GAL4/FLP combination? What is the frequency for other loci using the same Gal4/FLP combo and for the same locus using different Gal4s? (There is a hint of this answer in the legend of Figure 5*—*figure supplement 1; this should be addressed head on and quantified). Moreover, Figure 5 panel B seems to show quite a good number of cells that are green but not red and possibly a number of cells that are red but not green (e.g. in the calyx). How is this second population explained? This figure will benefit from higher resolution images and better quantification.*

To address this point, we have collected a number of new data sets that enabled us to quantitatively analyze the flipping efficiency of the FlpStop cassette 1) with different Gal4 lines (Figures 4 and 6) and 2) across different insertion loci (Figure 5 and Figure 5—figure supplement 1) and have rewritten the Results section accordingly (subsection “Labeling by the fluorescent reporter indicates efficient, cell type-specific cassette inversion”).

Flipping using different Gal4 lines:

We analyzed the flipping efficiency of the *cac^FlpStop^* allele using Gal4 lines that label the visual neurons Mi1, Tm1, T4 (Figure 4), and Tm3 (Figure 6) to drive Flp as well as GCaMP6f, which we used as a marker of the cells in the Gal4 pattern. These driver lines were chosen because they labeled many neurons within a single brain and because the cell bodies were spaced out enough from one another to enable quantification on a large scale in a confocal volume (Materials and methods). We quantified the fraction of GCaMP6f-positive cells that were also tdTomato-positive. These experiments strongly support our claim that the flipping efficiency is very high and will be suitable for many experimental goals. For Mi1, Tm1, and Tm3 greater than 90% of GCaMP6f-positive cells are also labeled by tdTomato. The T4 driver line is an extremely weak driver line (Fisher et al. 2015a), but we still observed 70% flipping.

Flipping efficiency across loci:

We have also examined the flipping efficiency of each of the FlpStop alleles. We used the Mi1 driver line to express Flp and the marker mCD8::GFP and quantified the proportion of GFP-positive cells that were also tdTomato-positive (Figure 5). Inversion efficiency was very high across many alleles: ~95% for *cac, ChAT, para, Rdl*, and *Shal*, and 75% for *ap*. We did not observe tdTomato-positive Mi1 cells using the *Gad1* and *VGlut* FlpStop alleles. However, using the pan-neuronal driver *nsyb-Gal4*, these alleles express tdTomato in a number of different cells (Figure 5—figure supplement 1). We believe that as previously reported, genome position effects on the expression of *UAS* transgenes (Markstein et al. 2008) resulted in the lack of tdTomato in Mi1. As demonstrated by tdTomato expression with *nsyb-Gal4*, the *Gad1* and *VGlut* FlpStop alleles invert well.

Are cells ever tdTomato-positive but not GFP-positive? The quantification we have now performed allowed us to address, across a number of Gal4 lines and FlpStop alleles, whether “red but not green cells” exist. The proportion of tdTomato-positive neurons with no detectable GFP staining was less than 0.01% for every genotype analyzed. Differences in the relative intensity of the tdTomato and GFP signals account for the appearance of a much larger fraction in images overlaying these channels.

*3) The advantages and disadvantages of FlpStop should be thoroughly discussed. With regard to the utility of the FlpStop toolkit to a potential end-user: a clear constraint is its dependence on the MiMIC insertion lines. The authors assert that MiMIC insertions "allow access to the coding introns of approximately 46% of neuronal genes" (subsection “FlpStop enables a diversity of applications”, first paragraph) and that "FlpStop alleles can be created that target any desired genomic locus using CRISPR/Cas9." The first statement seems likely to be a serious overestimate (using the authors methodology, I come up with MIMIC insertions into 23% of "neuronal genes").*

We thank the reviewers for their careful attention. We have reassessed our analysis and discovered that there was in fact an error in the code that led to an overestimation of the number of neuronal genes with MiMIC insertions in coding introns. We have redone the analysis and now find a number in line with that stated by the reviewers: 24% of neuronal genes. We have corrected this in the Discussion section (subsection “Targeting additional genetic loci using FlpStop”).

*While the second assertion is true, it would be more compelling if the manuscript presented and validated a FlpStop construct that was CRISPR/Cas ready if such a reagent has been made.*

We now describe a CRISPR-HDR-compatible FlpStop construct in the manuscript (Figure 1—figure supplement 2 and subsection “Targeting additional genetic loci using FlpStop”) and will deposit the plasmid at Addgene along with the MiMIC-compatible version.

*The need for a null allele for all non-X-linked genes should also be addressed. The general possibility of using the FlpStopD insertion itself is not discussed, but has the obvious disadvantage of unveiling second site mutations on the same chromosome. What other general strategies (e.g. deficiencies) exist, and how useful might they be?*

We have revised the Discussion section to explicitly address the requirement for a null allele for non-X-chromosome genes. This section reads:

“However, like MARCM, FlpStop requires a null allele of the gene of interest, though for FlpStop, this issue can be avoided for genes on the X chromosome by use of hemizygous males (Figures 6 and 7). […] This includes the disrupting FlpStop allele of the same gene and some deficiencies. If an established null does not already exist for a gene of interest, MiMIC insertions in the disrupting orientation can often provide a null allele and CRISPR-Cas9 technology enables the straightforward production of null alleles by creating targeted mutations within an essential exon (Venken et al. 2011; Bassett et al. 2013).”

*4) Subsection “FlpStop can produce conditional null alleles”. This section demonstrates at length that the FlpStop cassettes can do what MiMIC cassettes have already been shown to do – namely disrupt gene expression. While it is necessary to show that FlpStop works, this information would be of greater interest if it compared the efficacy of gene disruption using FlpStop with that of a MiMIC insertion in the disrupting orientation. One design difference between the MiMIC and FlpStop constructs is that the latter has transcriptional terminators, and it would be interesting to know if they are actually useful (i.e. do FlpStopDs reduce transcript levels more than the original MiMICs, when the latter are in a disrupting orientation?)*

The extent to which transcriptional terminators contribute to the mutagenicity of the FlpStop cassette is an interesting question, and one we have explored to some extent. In particular, when testing an older version of FlpStop, we inserted into *Rdl^MI02620^* two different cassettes that were identical, other than the presence of 1 terminator (SV40) or 2 terminators (SV40 and Tubα1). The second cassette was more mutagenic, as assessed using complementation testing and scoring for lethality. Thus, the terminators do contribute to disruption efficiency. However, the comparison between the MiMIC and FlpStop cassettes cannot be used to further address the role of transcriptional terminators. The MiMIC cassette contains the SV40 terminator following the EGFP as well as the terminator signals that follow the *yellow* marker in the same orientation as the gene trap (i.e. the splice acceptor and stop codons) (Venken et al. 2011). The FlpStop cassette contains an SV40 terminator and a Tubα1 terminator. When the direct comparison can be made between FlpStop disrupting alleles and MiMIC inserts in the disrupting orientation (*Rdl*, for example) both function as null alleles. As both cassettes contain multiple transcriptional terminators, interpreting their relative disruption efficacy is challenging.

We believe strongly that the more critical point of the data in this section (complementation testing) is that it serves to demonstrate that the non-disrupting orientation of the FlpStop cassette is inert. We have rephrased our writing to make this clearer (subsections “FlpStop can produce conditional null alleles”, first paragraph and “The current FlpStop tool kit”, last paragraph). In fact, an earlier generation of the FlpStop cassette had to be modified to exclude *mini-white* as it had mutagenic effects independent of the orientation of the SA-STOP. We speculate that this may have been due to the transitional terminator present within the *mini-white* gene.